# Variational Learning ISTA

**Fabio Valerio Massoli**                                            *fmassoli@qti.qualcomm.com*
*Qualcomm AI Research**

**Christos Louizos**                                               *clouizos@qti.qualcomm.com*
*Qualcomm AI Research**

**Arash Behboodi**                                                *abehboodi@qti.qualcomm.com*
*Qualcomm AI Research**

**Reviewed on OpenReview:** *https://openreview.net/forum?id=AQkOUsituG*

## Abstract

Compressed sensing combines the power of convex optimization techniques with a sparsity-inducing prior on the signal space to solve an underdetermined system of equations. For many problems, the sparsifying dictionary is not directly given, nor its existence can be assumed. Besides, the sensing matrix can change across different scenarios. Addressing these issues requires solving a sparse representation learning problem, namely dictionary learning, taking into account the epistemic uncertainty of the learned dictionaries and, finally, jointly learning sparse representations and reconstructions under varying sensing matrix conditions. We address both concerns by proposing a variant of the LISTA architecture. First, we introduce Augmented Dictionary Learning ISTA (A-DLISTA), which incorporates an augmentation module to adapt parameters to the current measurement setup. Then, we propose to learn a distribution over dictionaries via a variational approach, dubbed Variational Learning ISTA (VLISTA). VLISTA exploits A-DLISTA as the likelihood model and approximates a posterior distribution over the dictionaries as part of an unfolded LISTA-based recovery algorithm. As a result, VLISTA provides a probabilistic way to jointly learn the dictionary distribution and the reconstruction algorithm with varying sensing matrices. We provide theoretical and experimental support for our architecture and show that our model learns calibrated uncertainties.

## 1 Introduction

By imposing a prior on the signal structure, compressed sensing solves underdetermined inverse problems. Canonical examples of signal structure and sensing medium are sparsity and linear inverse problems. Compressed sensing aims at reconstructing an unknown signal of interest, $s \in \mathbb{R}^n$, from a set of linear measurements, $y \in \mathbb{R}^m$, acquired by means of a linear transformation, $\Phi \in \mathbb{R}^{m \times n}$ where $m < n$. Due to the underdetermined nature of the problem, $s$ is typically assumed to be sparse in a given basis. Hence, $s = \Psi x$, where $\Psi \in \mathbb{R}^{n \times b}$ is a matrix whose columns represent the sparsifying basis vectors, and $x \in \mathbb{R}^b$ is the sparse representation of $s$. Therefore, given noiseless observations $y = \Phi s$, of an unknown signal, $s = \Psi x$, we seek to solve the LASSO problem:

$$\underset{x}{\operatorname{argmin}} \|y - \Phi\Psi x\|_2^2 + \rho\|x\|_1 \tag{1}$$

where $\rho$ is a constant scalar controlling the sparsifying penalty. Iterative algorithms, such as Iterative Soft-Thresholding Algorithm (ISTA) (Daubechies et al., 2004), represent a popular approach to solving

---

*Qualcomm AI Research is an initiative of Qualcomm Technologies, Inc.

such problems. A number of studies have been conducted to improve compressed sensing solvers. A typical approach involves unfolding iterative algorithms as layers of neural networks and learning parameters end-to-end (Gregor & LeCun, 2010). Such ML algorithms are typically trained by minimizing the reconstruction objective:

$$\mathcal{L}(\boldsymbol{x}, \hat{\boldsymbol{x}}_T) = \mathbb{E}_{\boldsymbol{x} \sim \mathcal{D}}[\|\boldsymbol{x} - \hat{\boldsymbol{x}}_T\|_2^2] \tag{2}$$

where the expected value is taken over data sampled from $\mathcal{D}$, and the subscript "$T$" refers to the last step, or layer, of the unfolded model. Variable sensing matrices and unknown sparsifying dictionaries are some of the main challenges of data-driven approaches. By learning a dictionary and including it in optimization iterations, the work in Aberdam et al. (2021); Schnoor et al. (2022) aims to address these issues. However, data samples might not have exact sparse representations, so no ground truth dictionary is available. The issue can be more severe for heterogeneous datasets where the dictionary choice might vary from one sample to another. A real-world example would be the channel estimation problem in a Multi-Input Multi-Output (MIMO) mmwave wireless communication system (Rodríguez-Fernández et al., 2018). Such a problem can be cast as an inverse problem of the form $\boldsymbol{y} = \boldsymbol{\Phi\Psi x}$ and solved using compressive sensing techniques. The sensing matrix, $\boldsymbol{\Phi}$, represents the so-called *beamforming matrix* while the dictionary, $\boldsymbol{\Psi}$, represents the sparsifying basis for the wireless channel itself. Typically, $\boldsymbol{\Phi}$ changes from one set of measurements to the next and the channel model might require different basis vectors across time. Adaptive acquisition is another example of application: in MRI image reconstruction, the acquisition step can be adaptive. Here, the sensing matrix is sampled from a known distribution to reconstruct the signal. Therefore, given the adaptive nature of the process, each data sample is characterized by a different $\boldsymbol{\Phi}$ (Bakker et al., 2020; Yin et al., 2021).

**Our Contribution**   A principled approach to this problem would be to leverage a Bayesian framework and define a distribution over the dictionaries with proper uncertainty quantification. We follow two steps to accomplish this goal. First, we introduce Augmented Dictionary Learning ISTA (A-DLISTA), an augmented version of the Learning Iterative Soft-Thresholding Algorithm (LISTA)-like model, capable of adapting some of its parameters to the current measurement setup. We theoretically motivate its design and empirically prove its advantages compared to other non-adaptive LISTA-like models in a non-static measurement scenario, i.e., considering varying sensing matrices. Finally, to learn a distribution over dictionaries, we introduce Variational Learning ISTA (VLISTA), a variational formulation that leverages A-DLISTA as the likelihood model. VLISTA refines the dictionary iteratively after each iteration based on the outcome of the previous layer. Intuitively, our model can be understood as a form of a recurrent variational autoencoder, e.g., Chung et al. (2015), where at each iteration of the algorithm we have an approximate posterior distribution over the dictionaries conditioned on the outcome of the previous iteration. Moreover, VLISTA provides uncertainty estimation to detect Out-Of-Distribution (OOD) samples. We train A-DLISTA using the same objective as in Equation 2 while for VLISTA we maximize the ELBO (Equation 15). We refer the reader to Appendix D for the detailed derivation of the ELBO. Behrens et al. (2021) proposed an augmented version of LISTA, termed Neurally Augmented ALISTA (NALISTA). However, there are key differences with A-DLISTA. In contrast to NALISTA, our model adapts some of its parameters to the current sensing matrix and learned dictionary. Hypothetically, NALISTA could handle varying sensing matrices. However, that comes at the price of solving, for each datum, the inner optimization step to evaluate the $\mathbf{W}$ matrix. Finally, while NALISTA uses an LSTM as augmentation network, A-DLISTA employs a convolutional neural network (shared across all layers). Such a difference reflects the type of dependencies between layers and input data that the networks try to model. We report in Appendix B and Appendix C detailed discussions about the theoretical motivation and architectural design for A-DLISTA.

Our work's main contributions can be summarized as follows:

- We design an augmented version of a LISTA-like type of model, dubbed A-DLISTA, that can handle non-static measurement setups, i.e. per-sample sensing matrices, and adapt parameters to the current data instance.

- We propose VLISTA that learns a distribution over sparsifying dictionaries. The model can be interpreted as a Bayesian LISTA model that leverages A-DLISTA as the likelihood model.

- VLISTA adapts the dictionary to optimization dynamics and therefore can be interpreted as a hierarchical representation learning approach, where the dictionary atoms gradually permit more refined signal recovery.

- The dictionary distributions can be used successfully for out-of-distribution sample detection.

The remaining part of the paper is organized as follows. In section 2 we briefly report related works relevant to the current research. In section 3 and section 4 we introduce some background notions and details of our model formulations, respectively. Datasets, baselines, and experimental results are described in section 5. Finally, we draw our conclusion in section 6.

## 2 Related Works

In compressed sensing, recovery algorithms have been extensively analyzed theoretically and numerically (Foucart & Rauhut, 2013). One of the most prominent approaches is using iterative algorithms, such as: ISTA (Daubechies et al., 2004), Approximate message passing (AMP) (Donoho et al., 2009) Orthogonal Matching Pursuit (OMP) (Pati et al., 1993; Davis et al., 1994), and Iterative Hard-Thresholding Algorithm (IHTA) (Blumensath & Davies, 2009). These algorithms have associated hyperparameters, including the number of iterations and soft threshold, which can be adjusted to better balance performance and complexity. With unfolding iterative algorithms as layers of neural networks, these parameters can be learned in an end-to-end fashion from a dataset see, for instance, some variants Zhang & Ghanem (2018); Metzler et al. (2017); yang et al. (2016); Borgerding et al. (2017); Sprechmann et al. (2015). In previous studies by Zhou et al. (2009; 2012), a non-parametric Bayesian method for dictionary learning was presented. The authors focused on a fully Bayesian joint compressed sensing inversion and dictionary learning, where the dictionary atoms were drawn and fixed beforehand. Bayesian compressive sensing (BCS) (Ji et al., 2008) uses relevance vector machines (RVMs) (Tipping, 2001) and a hierarchical prior to model distributions of each entry. This line of work quantifies the uncertainty of recovered entries while assuming a fixed dictionary. Our current work differs by accounting for uncertainty in the unknown dictionary by defining a distribution over it. Learning ISTA was initially introduced by Gregor & LeCun (2010). Since then, many works have followed, including those by Behrens et al. (2021); Liu et al. (2019); Chen et al. (2021); Wu et al. (2020). These subsequent works provide guidelines for improving LISTA, for example, in convergence, parameter efficiency, step size and threshold adaptation, and overshooting. However, they assume fixed and known sparsifying dictionaries and sensing matrices. Researches by Aberdam et al. (2021); Behboodi et al. (2022); Schnoor et al. (2022) have explored ways to relax these assumptions, including developing models that can handle varying sensing matrices and learn dictionaries. The authors in Schnoor et al. (2022); Behboodi et al. (2022) provide an architecture that can both incorporate varying sensing matrices and learn dictionaries. However, their focus is on the theoretical analysis of the model. Furthermore, there are theoretical studies on the convergence and generalization of unfolded networks, see for example: Giryes et al. (2018); Pu et al. (2022); Aberdam et al. (2021); Pu et al. (2022); Chen et al. (2018); Behboodi et al. (2022); Schnoor et al. (2022). Our paper builds on these ideas by modelling a distribution over dictionaries and accounting for epistemic uncertainty. Previous studies have explored theoretical aspects of unfolded networks, such as convergence and generalization, and we contribute to this body of research by considering the impact of varying sensing matrices and dictionaries. The framework of variational autoencoders (VAEs) enables the learning of a generative model through latent variables (Kingma & Welling, 2013; Rezende et al., 2014). When there are data-sample-specific dictionaries in our proposed model, it reminisces extensions of VAEs to the recurrent setting (Chung et al., 2015; 2016), which assumes a sequential structure in the data and imposes temporal correlations between the latent variables. Additionally, there are connections and similarities to Markov state-space models, such as the ones described in Krishnan et al. (2017). By using global dictionaries in VLISTA, the model becomes a variational Bayesian Recurrent Neural Network. Variational Bayesian neural networks were first introduced in Blundell et al. (2015), with independent priors and variational posteriors for each layer. This work has been extended to recurrent settings in Fortunato et al. (2019). The main difference between these works and our setting is the prior and variational posterior. At each step, the prior and variational posterior are conditioned on previous steps instead of being fixed across steps.

## 3 Background

### 3.1 Sparse linear inverse problems

We consider linear inverse problems of the form: $\boldsymbol{y} = \boldsymbol{\Phi}\boldsymbol{s}$, where we have access to a set of linear measurements $\boldsymbol{y} \in \mathbb{R}^m$ of an unknown signal $\boldsymbol{s} \in \mathbb{R}^n$, acquired through the forward operator $\boldsymbol{\Phi} \in \mathbb{R}^{m \times n}$. Typically, in compressed sensing literature, $\boldsymbol{\Phi}$ is called the sensing, or measurements, matrix and it represents an underdetermined system of equations for $m < n$. The problem of reconstructing $\boldsymbol{s}$ from $(\boldsymbol{y}, \boldsymbol{\Phi})$ is ill-posed due to the shape of the forward operator. To uniquely solve for $\boldsymbol{s}$, the signal is assumed to admit a sparse representation, $\boldsymbol{x} \in \mathbb{R}^b$, in a given basis, $\{e_i \in \mathbb{R}^n\}_{i=0}^b$. The $e_i$ vectors are called *atoms* and are collected as the columns of a matrix $\boldsymbol{\Psi} \in \mathbb{R}^{n \times b}$ termed the *sparsifying dictionary*. Therefore, the problem of estimating $\boldsymbol{s}$, given a limited number of observations $\boldsymbol{y}$ through the operator $\boldsymbol{\Phi}$, is translated to a sparse recovery problem: $\boldsymbol{x}^* = \arg\min_{\boldsymbol{x}} \|\boldsymbol{x}\|_0$ s.t. $\boldsymbol{y} = \boldsymbol{\Phi}\boldsymbol{\Psi}\boldsymbol{x}$. Given that the $l_0$ pseudo-norm requires solving an NP-hard problem, the $l_1$ norm is used instead as a convex relaxation of the problem. A proximal gradient descent-based approach for solving the problem yields the ISTA algorithm (Daubechies et al., 2004; Beck & Teboulle, 2009):

$$\boldsymbol{x}_t = \eta_{\theta_t}\left(\boldsymbol{x}_{t-1} + \gamma_t(\boldsymbol{\Phi}\boldsymbol{\Psi})^T(\boldsymbol{y} - \boldsymbol{\Phi}\boldsymbol{\Psi}\boldsymbol{x}_{t-1})\right), \tag{3}$$

where $t$ is the index of the current iteration, $\boldsymbol{x}_t$ ($\boldsymbol{x}_{t-1}$) is the reconstructed sparse vector at the current (previous) layer, and $\theta_t$ and $\gamma_t$ are the *soft-threshold* and *step size* hyperparameters, respectively. Specifically, $\theta_t$ characterizes the *soft-threshold function* given by: $\eta_{\theta_t}(\boldsymbol{x}) = \text{sign}(\boldsymbol{x})(|\boldsymbol{x}| - \theta_t)_+$. In the ISTA formulation, those two parameters are shared across all the iterations: $\gamma_t, \theta_t \to \gamma, \theta$. In what follows, we use the terms "layers" and "iterations" interchangeably when describing ISTA and its variations.

### 3.2 LISTA

LISTA (Gregor & LeCun, 2010) is an unfolded version of the ISTA algorithm in which each iteration is parametrized by learnable matrices. Specifically, LISTA reinterprets Equation 3 as defining the layer of a feed-forward neural network implemented as $S_{\theta_t}(\boldsymbol{V}_t\boldsymbol{x}_{t-1} + \boldsymbol{W}_t\boldsymbol{y})$ where $\boldsymbol{V}_t, \boldsymbol{W}_t$ are learnt from a dataset. In that way, those weights implicitly contain information about $\boldsymbol{\Phi}$ and $\boldsymbol{\Psi}$ that are assumed to be fixed. As LISTA, also its variations, e.g., Analytic LISTA (ALISTA) (Liu et al., 2019), NALISTA (Behrens et al., 2021) and HyperLISTA (Chen et al., 2021), require similar constraints such as a fixed dictionary and sensing matrix to reach the best performance. However, there are situations where one or none of the conditions are met (see examples in section 1).

## 4 Method

### 4.1 Augmented Dictionary Learning ISTA (A-DLISTA)

To deal with situations where $\boldsymbol{\Psi}$ is unknown and $\boldsymbol{\Phi}$ is changing across samples, one can unfold the ISTA algorithm and re-parametrize the dictionary as a learnable matrix. Such an algorithm is termed Dictionary Learning ISTA (DLISTA) (Pezeshki et al., 2022; Behboodi et al., 2022; Aberdam et al., 2021) and, similarly to Equation 3, each layer is formulated as:

$$\boldsymbol{x}_t = \eta_{\theta_t}\left(\boldsymbol{x}_{t-1} + \gamma_t(\boldsymbol{\Phi}\boldsymbol{\Psi}_t)^T(\boldsymbol{y} - \boldsymbol{\Phi}\boldsymbol{\Psi}_t\boldsymbol{x}_{t-1})\right), \tag{4}$$

with one last linear layer mapping $\boldsymbol{x}$ to the reconstructed signal $\boldsymbol{s}$. The model can be trained end-to-end to learn all $\theta_t, \gamma_t, \boldsymbol{\Psi}_t$. Differently from ISTA (Equation 3), DLISTA (Equation 4) learns a dictionary specific for each layer, indicated by the subscript "$t$". The model can be trained end-to-end to learn all $\theta_t, \gamma_t, \boldsymbol{\Psi}_t$. The base model is similar to (Behboodi et al., 2022; Aberdam et al., 2021). However, it requires additional changes. Consider the $t$-th layer of DLISTA with the varying sensing matrix $\boldsymbol{\Phi}^k$ and define the following

parameters:

$$\tilde{\mu}(t, \mathbf{\Phi}^k) := \max_{1 \le i \ne j \le N} \left| ((\mathbf{\Phi}^k \mathbf{\Psi}_t)_i)^\top (\mathbf{\Phi}^k \mathbf{\Psi}_t)_j \right| \tag{5}$$

$$\tilde{\mu}_2(t, \mathbf{\Phi}^k) := \max_{1 \le i,j \le N} \left| ((\mathbf{\Phi}^k \mathbf{\Psi}_t)_i)^\top (\mathbf{\Phi}^k (\mathbf{\Psi}_t - \mathbf{\Psi}_o))_j \right| \tag{6}$$

$$\delta(\gamma, t, \mathbf{\Phi}^k) := \max_i \left| 1 - \gamma \left\| (\mathbf{\Phi}^k \mathbf{\Psi}_t)_i \right\|_2^2 \right| \tag{7}$$

where $\tilde{\mu}$ is the *mutual coherence* of $\mathbf{\Phi}^k \mathbf{\Psi}_t$ (Foucart & Rauhut, 2013, Chapter 5) and $\tilde{\mu}_2$ is closely connected to *generalized mutual coherence* (Liu et al., 2019). However, in contrast to the generalized mutual coherence, $\tilde{\mu}_2$ includes the diagonal inner product for $i = j$. Finally, $\delta(\cdot)$ is reminiscent of the restricted isometry property (RIP) constant (Foucart & Rauhut, 2013), a key condition for many recovery guarantees in compressed sensing. When the columns of the matrix $\mathbf{\Phi}^k \mathbf{\Psi}_t$ are normalized, the choice of $\gamma = 1$ yields $\delta(\gamma, t, \mathbf{\Phi}^k) = 0$. The following proposition provides conditions on each layer to improve the reconstruction error.

**Proposition 4.1.** *Suppose that $\boldsymbol{y}^k = \mathbf{\Phi}^k \mathbf{\Psi}_o \boldsymbol{x}_*$, where $\boldsymbol{x}_*$ is the ground truth sparse vector with support $supp(\boldsymbol{x}_*) = S$, and $\mathbf{\Psi}_o$ is the ground truth dictionary. For DLISTA iterations given as*

$$\boldsymbol{x}_t = \eta_{\theta_t} \left( \boldsymbol{x}_{t-1} + \gamma_t (\mathbf{\Phi}^k \mathbf{\Psi}_t)^T (\boldsymbol{y}^k - \mathbf{\Phi}^k \mathbf{\Psi}_t \boldsymbol{x}_{t-1}) \right), \tag{8}$$

*we have:*

1. *If for all $t$, the pairs $(\theta_t, \gamma_t, \mathbf{\Psi}_t)$ satisfy*

$$\gamma_t \left( \tilde{\mu} \left\| \boldsymbol{x}_* - \boldsymbol{x}_{t-1} \right\|_1 + \tilde{\mu}_2 \left\| \boldsymbol{x}_* \right\|_1 \right) \le \theta_t, \tag{9}$$

   *then there is no false positive in each iteration. In other words, for all $t$, we have $supp(\boldsymbol{x}_t) \subseteq supp(\boldsymbol{x}_*)$.*

2. *Assuming that the conditions of the last step hold, then we get the following bound on the error:*

$$\left\| \boldsymbol{x}_t - \boldsymbol{x}_* \right\|_1 \le \left( \delta(\gamma_t) + \gamma_t \tilde{\mu}(|S| - 1) \right) \left\| \boldsymbol{x}_{t-1} - \boldsymbol{x}_* \right\|_1 + \gamma_t \tilde{\mu}_2 |S| \left\| \boldsymbol{x}_* \right\|_1 + |S| \theta_t.$$

We provide the derivation of Proposition 4.1 together with additional theoretical results in Appendix B. Proposition 4.1 provides insights about the choice of $\gamma_t$ and $\theta_t$, and also suggests that $(\delta(\gamma_t) + \gamma_t \tilde{\mu}(|S| - 1))$ needs to be smaller than one to reduce the error at each step.

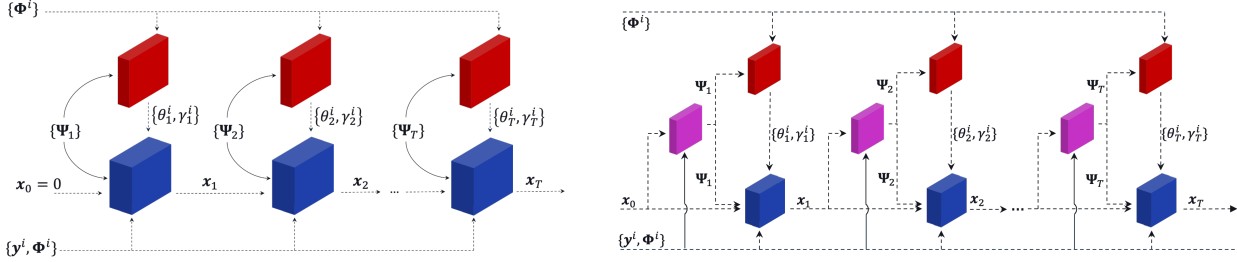

**Figure 1: Models architectures**. **Left:** A-DLISTA architecture. Each blue block represents a single ISTA-like iteration parametrized by the dictionary $\mathbf{\Psi}_t$, the threshold and step size $\{\theta_t^i, \gamma_t^i\}$. The red blocks represent the augmentation network (with shared parameters across layers) that adapts $\{\theta_t^i, \gamma_t^i\}$ for layer $t$ based on the dictionary $\mathbf{\Psi}_t$ and the current measurement setup $\mathbf{\Phi}^i$ for the $i-$th data sample. **Right:** VLISTA (inference) architecture. The red and blue blocks correspond to the same operations as for A-DLISTA. The pink blocks represent the posterior model used to refine the dictionary based on input data $\{\boldsymbol{y}^i, \mathbf{\Phi}^i\}$ and the sparse vector reconstructed at layer $t$, $\mathbf{x}_t$.

Upon examining Proposition 4.1, it becomes evident that $\gamma_t$ and $\theta_t$ play a key role in the convergence of the algorithm. However, there is a trade-off to consider when making these choices. For instance, suppose we set

| **Algorithm 1** Augmented Dictionary Learning ISTA (A-DLISTA) - Inference Algorithm | **Algorithm 2** Variational Learning ISTA (VLISTA) - Inference Algorithm |
|---|---|
| **Require:** $\mathcal{D} = \{(\boldsymbol{y}^i, \boldsymbol{\Phi}^i)\}_{i=0}^{N-1}$ - the sensing matrix changes across samples; augmentation model $f_\Theta$ 
 $\quad \boldsymbol{x}_0^i \leftarrow 0$ 
 $\quad$ **for** $t = 1, \ldots, T$ **do** 
 $\quad\quad (\theta_t^i, \gamma_t^i) \leftarrow f_\Theta(\boldsymbol{\Phi}^i, \boldsymbol{\Psi}_t) \quad\quad \triangleright$ Augmentation step 
 $\quad\quad g \leftarrow \boldsymbol{y}^i - (\boldsymbol{\Phi}^i \boldsymbol{\Psi}_t)^T \boldsymbol{\Phi}^i \boldsymbol{\Psi}_t \boldsymbol{x}_{t-1}^i$ 
 $\quad\quad u \leftarrow \boldsymbol{x}_{t-1}^i + \gamma_t^i g$ 
 $\quad\quad \boldsymbol{x}_t^i = \eta_{\theta_t^i}(u)$ 
 $\quad$ **end for** 
 $\quad$ **return** $\boldsymbol{x}_T^i$; | **Require:** $\mathcal{D} = \{(\boldsymbol{y}^i, \boldsymbol{\Phi}^i)\}_{i=0}^{N-1}$ - the sensing matrix changes across samples; augmentation model $f_\Theta$; posterior model $f_\phi$ 
 $\quad \boldsymbol{x}_0^i \leftarrow 0$ 
 $\quad$ **for** $t = 1, \ldots, T$ **do** 
 $\quad\quad (\boldsymbol{\mu}_t, \boldsymbol{\sigma^2}_t) \leftarrow f_\phi(\boldsymbol{x}_{t-1}^i, \boldsymbol{y}^i, \boldsymbol{\Phi}^i) \quad\quad \triangleright$ Posterior parameters estimation 
 $\quad\quad \boldsymbol{\Psi}_t \sim \mathcal{N}(\boldsymbol{\Psi}_t \mid \boldsymbol{\mu}_t, \boldsymbol{\sigma^2}_t) \quad \triangleright$ Dictionary sampling 
 $\quad\quad (\theta_t^i, \gamma_t^i) \leftarrow f_\Theta(\boldsymbol{\Phi}^i, \boldsymbol{\Psi}_t) \quad \triangleright$ Augmentation step 
 $\quad\quad g \leftarrow \boldsymbol{y}^i - (\boldsymbol{\Phi}^i \boldsymbol{\Psi}_t)^T \boldsymbol{\Phi}^i \boldsymbol{\Psi}_t \boldsymbol{x}_{t-1}^i$ 
 $\quad\quad u \leftarrow \boldsymbol{x}_{t-1}^i + \gamma_t^i g$ 
 $\quad\quad \boldsymbol{x}_t^i = \eta_{\theta_t^i}(u)$ 
 $\quad$ **end for** 
 $\quad$ **return** $\boldsymbol{x}_T^i$; |

$\theta_t$ and decrease $\gamma_t$. In that case, we may ensure good support selection, but it could also increase $\delta(\gamma_t)$. In situations where the sensing matrix remains fixed, the network can possibly learn optimal choices through end-to-end training. However, when the sensing matrix $\boldsymbol{\Phi}$ differs across various data samples (i.e., $\boldsymbol{\Phi} \to \boldsymbol{\Phi}^i$), it is no longer guaranteed that there exists a unique choice of $\gamma_t$ and $\theta_t$ for all $\boldsymbol{\Phi}^i$. Since these parameters can be determined when $\boldsymbol{\Phi}$ and $\boldsymbol{\Psi}_t$ are fixed, we suggest utilizing an augmentation network to determine $\gamma_t$ and $\theta_t$ from each pair of $\boldsymbol{\Phi}^i$ and $\boldsymbol{\Psi}_t$. For a more thorough theoretical analysis, please refer to Appendix B.

We show in Figure 1 (left plot) the resulting A-DLISTA model. At each layer, A-DLISTA performs two basic operations, namely, soft-threshold (blue blocks in Figure 1) and augmentation (red blocks in Figure 1). The former represents an ISTA-like iteration parametrized by the set of weights $\{\boldsymbol{\Psi}_t, \theta_t^i, \gamma_t^i\}$, whilst the latter is implemented using a convolutional neural network. As shown in the figure, the augmentation network takes as input the sensing matrix for the given data sample, $\boldsymbol{\Phi}^i$, together with the dictionary learned at the layer for which the augmentation model will generate the $\theta_t^i$ and $\gamma_t^i$ parameters: $(\theta_t^i, \gamma_t^i) = f_\Theta(\boldsymbol{\Phi}^i, \boldsymbol{\Psi}_t)$, where $\boldsymbol{\Theta}$ are the augmentation models' trainable parameters. Through such an operation, A-DLISTA adapts the soft-threshold and step size of each layer to the current data sample. The inference algorithmic description of A-DLISTA is given in Algorithm 1. We report more details about the augmentation network in the supplementary materials (Appendix C).

## 4.2 Variational Learning ISTA

Although A-DLISTA possesses adaptivity to data samples, it still assumes the existence of a ground truth dictionary. We relax such a hypothesis by defining a probability distribution over $\boldsymbol{\Psi}_t$ and formulating a variational approach, titled VLISTA, to solve the dictionary learning and sparse recovery problems jointly. To forge our variational framework whilst retaining the helpful adaptivity property of A-DLISTA, we re-interpret the soft-thresholding layers of the latter as part of a likelihood model defining the output mean for the reconstructed signal. Given its recurrent-like structure (Chung et al., 2015), we equip VLISTA with a conditional trainable prior where the condition is given by the dictionary sampled at the previous iteration. Therefore, the full model comprises three components, namely, the conditional prior $p_\xi(\cdot)$, the variational posterior $q_\phi(\cdot)$, and the likelihood model, $p_\Theta(\cdot)$. All components are parametrized by neural networks whose outputs represent the parameters for the underlying probability distribution. In what follows, we describe more in detail the various building blocks of the VLISTA model.

### 4.2.1 Prior Model

The conditional prior, $p_\xi(\boldsymbol{\Psi}_t \mid \boldsymbol{\Psi}_{t-1})$, is modelled as a Gaussian distribution whose parameters are conditioned on the previously sampled dictionary. We implement $p_\xi(\cdot)$ as a neural network, $f_\xi(\cdot) = [f_{\xi_1}^\mu \circ g_{\xi_0}(\cdot), f_{\xi_2}^{\sigma^2} \circ g_{\xi_0}(\cdot)]$,

with trainable parameters $\xi = \{\xi_0, \xi_1, \xi_2\}$. The model's architecture comprises a shared convolutional block followed by two branches generating the Gaussian distribution's mean and standard deviation, respectively. Therefore, at layer $t$, the prior conditional distribution is given by: $p_\xi(\boldsymbol{\Psi}_t|\boldsymbol{\Psi}_{t-1}) = \prod_{i,j} \mathcal{N}(\boldsymbol{\Psi}_{t;i,j}|\mu_{t;i,j} = f_{\xi_1}^\mu(g_{\xi_0}(\boldsymbol{\Psi}_{t-1}))_{i,j}; \sigma_{t;i,j} = f_{\xi_2}^{\sigma^2}(g_{\xi_0}(\boldsymbol{\Psi}_{t-1}))_{i,j})$, where the indices $i, j$ run over the rows and columns of $\boldsymbol{\Psi}_t$. To simplify our expressions, we will abuse notation and refer to distributions like the former one as:

$$p_\xi(\boldsymbol{\Psi}_t|\boldsymbol{\Psi}_{t-1}) = \mathcal{N}(\boldsymbol{\Psi}_t|\boldsymbol{\mu}_t; \boldsymbol{\sigma}^2{}_t), \quad \text{where} \tag{10}$$
$$\boldsymbol{\mu}_t = f_{\xi_1}^\mu(g_{\xi_0}(\boldsymbol{\Psi}_{t-1})); \quad \boldsymbol{\sigma}^2{}_t = f_{\xi_2}^{\sigma^2}(g_{\xi_0}(\boldsymbol{\Psi}_{t-1}))$$

We will use the same type of notation throughout the rest of the manuscript to simplify formulas. The prior design allows for enforcing a dependence of the dictionary at iteration $t$ to the one sampled at the previous iteration. Thus allowing us to refine $\boldsymbol{\Psi}_t$ as the iterations proceed. The only exception is the prior imposed over the dictionary at $t = 1$, where there is no previously sampled dictionary. To handle this exception, we assume a standard Gaussian distributed $\boldsymbol{\Psi}_1$. The joint prior distribution over the dictionaries for VLISTA is given by:

$$p_\xi(\boldsymbol{\Psi}_{1:T}) = \mathcal{N}(\boldsymbol{\Psi}_1|\mathbf{0}; \mathbf{1}) \prod_{t=2}^{T} \mathcal{N}(\boldsymbol{\Psi}_t|\boldsymbol{\mu}_t; \boldsymbol{\sigma}^2{}_t) \tag{11}$$

where $\boldsymbol{\mu}_t$ and $\boldsymbol{\sigma}^2{}_t$ are defined in Equation 10.

### 4.2.2 Posterior Model

Similarly to the prior model, the variational posterior too is modelled as a Gaussian distribution parametrized by a neural network $f_\phi(\cdot) = [f_{\phi_1}^\mu \circ h_{\phi_0}(\cdot), f_{\phi_2}^{\sigma^2} \circ h_{\phi_0}(\cdot)]$ that outputs the mean and variance for the underlying probability distribution

$$q_\phi(\boldsymbol{\Psi}_t|\boldsymbol{x}_{t-1}, \boldsymbol{y}^i, \boldsymbol{\Phi}^i) = \mathcal{N}(\boldsymbol{\Psi}_t|\boldsymbol{\mu}_t; \boldsymbol{\sigma}^2{}_t), \quad \text{where} \tag{12}$$
$$\boldsymbol{\mu}_t = f_{\phi_1}^\mu(h_{\phi_0}(\boldsymbol{x}_{t-1}, \boldsymbol{y}^i, \boldsymbol{\Phi}^i)); \quad \boldsymbol{\sigma}^2{}_t = f_{\phi_2}^{\sigma^2}(h_{\phi_0}(\boldsymbol{x}_{t-1}, \boldsymbol{y}^i, \boldsymbol{\Phi}^i))$$

The posterior distribution for the dictionary, $\boldsymbol{\Psi}_t$, at layer $t$ is conditioned on the data, $\{\boldsymbol{y}^i, \boldsymbol{\Phi}^i\}$, as well as on the reconstructed signal at the previous layer, $\boldsymbol{x}_{t-1}$. Therefore, the joint posterior probability over the dictionaries is given by:

$$q_\phi(\boldsymbol{\Psi}_{1:T}|\boldsymbol{x}_{1:T}, \boldsymbol{y}^i, \boldsymbol{\Phi}^i) = \prod_{t=1}^{T} q_\phi(\boldsymbol{\Psi}_t|\boldsymbol{x}_{t-1}, \boldsymbol{y}^i, \boldsymbol{\Phi}^i) \tag{13}$$

When considering our selection of Gaussian distributions for our prior and posterior models, we prioritized computational and implementation convenience. However, it's important to note that our framework is not limited to this distribution family. As long as the distributions used are reparametrizable (Kingma & Welling, 2013), meaning that we can obtain gradients of random samples with respect to their parameters and we can evaluate and differentiate their density, VLISTA can support any flexible distribution family. This includes mixtures of Gaussians to incorporate heavier tails and distributions resulting from normalizing flows (Rezende & Mohamed, 2015).

### 4.2.3 Likelihood Model

The soft-thresholding block of A-DLISTA is at the heart of the reconstruction module. Similarly to the prior and posterior, the likelihood distribution is modelled as a Gaussian parametrized by the output of an A-DLISTA block. In particular, the network generates the mean vector for the Gaussian distribution while we treat the standard deviation as a tunable hyperparameter. By combining these elements, we can formulate the joint log-likelihood distribution as:

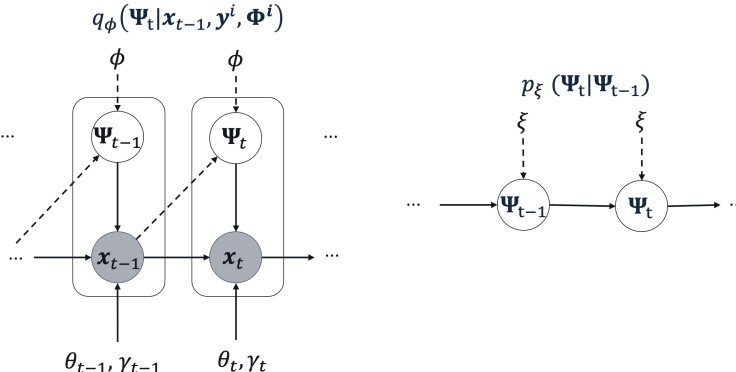

**Figure 2: VLISTA graphical model**. Dependencies on $\boldsymbol{y}^i$ and $\boldsymbol{\Phi}^i$ are factored out for simplicity. The sampling is done only based on the posterior $q_\phi(\boldsymbol{\Psi}_t|\boldsymbol{x}_{t-1}, \boldsymbol{y}^i, \boldsymbol{\Phi}^i)$. Dashed lines represent variational approximations.

$$\log p_\Theta(\boldsymbol{x}_{1:T} = \boldsymbol{x}^i_{gt}|\boldsymbol{\Psi}_{1:T}, \boldsymbol{y}^i, \boldsymbol{\Phi}^i) = \sum_{t=1}^{T} \log \mathcal{N}(\boldsymbol{x}^i_{gt}|\boldsymbol{\mu}_t, \boldsymbol{\sigma}^2{}_t), \quad \text{where} \tag{14}$$

$$\boldsymbol{\mu}_t = \text{A-DLISTA}(\boldsymbol{\Psi}_t, \boldsymbol{x}_{t-1}, \boldsymbol{y}^i, \boldsymbol{\Phi}^i; \Theta); \quad \boldsymbol{\sigma}^2{}_t = \delta$$

where $\delta$ is a hyperparameter of the network, $\boldsymbol{x}^i_{gt}$ represents the ground truth value for the underlying unknown sparse signal for the $i$-th data sample, $\Theta$ is the set of A-DLISTA's parameters, and $p_\Theta(\boldsymbol{x}_{1:T} = \boldsymbol{x}^i_{gt}|\cdot)$ represents the likelihood for the ground truth $x^i_{gt}$, at each time step $t \in [1, T]$, under the likelihood model given the current reconstruction. Note that in Equation 14 we use the same $\boldsymbol{x}^i_{gt}$ through the entire sequence $t \in [1, T]$. We report in Figure 1 (right plot) the inference architecture for VLISTA. It is crucial to note that during inference VLISTA does not require the prior model which is used for training only. Additionally, its graphical model and algorithmic description are shown in Figure 2 and Algorithm 2, respectively. We report further details concerning the architecture of the models and the objective function in the supplementary material (Appendix C and Appendix D).

### 4.2.4 Training Objective

Our approach involves training all VLISTA components in an end-to-end fashion. To accomplish that we maximize the Evidence Lower Bound (ELBO):

$$\text{ELBO} = \sum_{t=1}^{T} \mathbb{E}_{\boldsymbol{\Psi}_{1:t} \sim q_\phi(\boldsymbol{\Psi}_{1:t}|\boldsymbol{x}_{0:t-1}, \boldsymbol{D}^i)} \Big[\log p_\Theta(\boldsymbol{x}_t = \boldsymbol{x}^i_{gt}|\boldsymbol{\Psi}_{1:t}, \boldsymbol{D}^i)\Big] \tag{15}$$
$$- \sum_{t=2}^{T} \mathbb{E}_{\boldsymbol{\Psi}_{1:t-1} \sim q_\phi(\boldsymbol{\Psi}_{1:t-1}|\boldsymbol{x}_{t-1}, \boldsymbol{D}^i)} \Big[D_{KL}\Big(q_\phi(\boldsymbol{\Psi}_t|\boldsymbol{x}_{t-1}, \boldsymbol{D}^i) \parallel p_\xi(\boldsymbol{\Psi}_t|\boldsymbol{\Psi}_{t-1})\Big)\Big]$$
$$- D_{KL}\Big(q_\phi(\boldsymbol{\Psi}_1|\boldsymbol{x}_0, \boldsymbol{D}^i) \parallel p(\boldsymbol{\Psi}_1)\Big)$$

As we can see from Equation 15, the ELBO comprises three terms. The first term is the sum of expected log-likelihoods of the target signal at each time step. The second term is the sum of KL divergences between the approximate posterior and the prior at each time step. The third term is the KL divergence between the approximate posterior at the initial time step and a prior. In our implementation, we set to "$T$" the number of layers and initialize the input signal to zero.

To evaluate the likelihood contribution in Equation 15, we marginalize over dictionaries sampled from the posterior $q_\phi(\mathbf{\Psi}_{1:t}|\boldsymbol{x}_{0:t-1}, \boldsymbol{D}^i)$. In contrast, the last two terms in the equation represent the KL divergence contribution between the prior and posterior distributions. It's worth noting that the prior in the last term is not conditioned on the previously sampled dictionary, given that $p_\xi(\mathbf{\Psi}_1) \rightarrow p(\mathbf{\Psi}_1) = \mathcal{N}(\mathbf{\Psi}_1|\boldsymbol{0}; \boldsymbol{1})$ (refer to Equation 10 and Equation 11). We refer the reader to Appendix D for the derivstion of the ELBO.

## 5 Experiments

### 5.1 Datasets and Baselines

We evaluate our models' performance by comparing them against classical and ML-based baselines on three datasets: MNIST, CIFAR10, and a synthetic dataset. Concerning the synthetic dataset, we follow a similar approach as in Chen et al. (2018); Liu & Chen (2019); Behrens et al. (2021). However, in contrast to the mentioned works, we generate a different $\mathbf{\Phi}$ matrix for each datum by sampling i.i.d. entries from a standard Gaussian distribution. We generate the ground truth sparse signals by sampling the entries from a standard Gaussian and setting each entry to be non-zero with a probability of 0.1. We generate 5K samples and use 3K for training, 1K for model selection, and 1K for testing. Concerning the MNIST and CIFAR10, we train the models using the full images, without applying any crop. For CIFAR10, we gray-scale and normalize the images. We generate the corresponding observations, $\boldsymbol{y}^i$, by multiplying each sensing matrix with the ground truth image: $\boldsymbol{y}^i = \mathbf{\Phi}^i \boldsymbol{s}^i$. We compare the A-DLISTA and VLISTA models against classical and ML baselines. Our classical baselines use the ISTA algorithm, and we pre-compute the dictionary by either considering the canonical or the wavelet basis or using the SPCA algorithm. Our ML baselines use different unfolded learning versions of ISTA, such as LISTA. To demonstrate the benefits of adaptivity, we perform an ablation study on A-DLISTA by removing its augmentation network and making the parameters $\theta_t, \gamma_t$ learnable only through backpropagation. We refer to the non-augmented version of A-DLISTA as DLISTA. Therefore, for DLISTA, $\theta_t$ and $\gamma_t$ cannot be adapted to the specific input sensing matrix. Moreover, we consider BCS (Ji et al., 2008) as a specific Bayesian baseline for VLISTA. Finally, we conduct Out-Of-Distribution (OOD) detection experiments. We fixed the number of layers to three for all ML models to compare their performance. The classical baselines do not possess learnable parameters. Therefore, we performed an extensive grid search to find the best hyperparameters for them. More details concerning the training procedure and ablation studies can be found in Appendix D and Appendix F.

### 5.2 Synthetic Dataset

Regarding the synthetic dataset, we evaluate models performance by computing the median of the c.d.f. for the reconstruction NMSE (Figure 3). A-DLISTA's adaptivity appears to offer an advantage over other models. However, concerning VLISTA, we observe a drop in performance. Such a behaviour is consistent across experiments and can be attributed to a few factors. One possible reason for the drop in performance is the noise introduced during training due to the random sampling procedure used to generate the dictionary. Additionally, the amortization gap that affects all models based on amortized variational inference (Cremer et al., 2018) can also contribute to this effect. Despite this, VLISTA still performs comparably to BCS. Lastly, we note that ALISTA and NALISTA do not perform as well as other models. This is likely due to the optimization procedure these

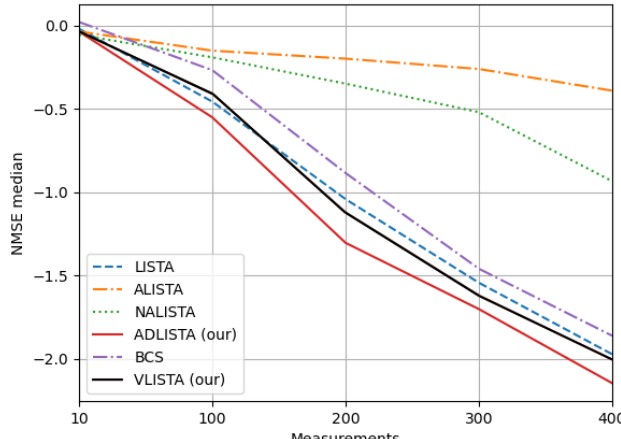

**Figure 3: NMSE's median**. The $y$-axes is in dB (the lower the better) for a different number of measurements ($x$-axes).

two models require to evaluate the weight matrix $\boldsymbol{W}$. The computation of the $\boldsymbol{W}$ matrix requires a fixed

sensing matrix, a condition not satisfied in the current setup. Regarding non-static measurements, we averaged across multiple $\boldsymbol{\Phi}^i$, thus obtaining a non-optimal $W$ matrix. To support our hypothesis, we report in Appendix F results considering a static measurement scenario for which ALISTA and NALISTA report very high performance.

## 5.3 Image Reconstruction - MNIST & CIFAR10

When evaluating different models on the MNIST and CIFAR10 datasets, we use the Structural Similarity Index Measure (SSIM) to measure their performance. As for the synthetic dataset, we experience strong instabilities in ALISTA and NALISTA training due to non-static measurement setup. Therefore, we do not provide results for these models. It is important to note that the poor performance of ALISTA and NALISTA is a result of our specific experiment setup, which differs from the static case considered in the formulation of these models. We refer to Appendix F for results using a static measurements scenario. By looking at the results in Table 1 and Table 2, we can draw similar conclusions as for the synthetic dataset. Additionally, we report results from three classical baselines (subsection 5.1). Among non-Bayesian models, A-DLISTA shows the best results. Furthermore, by comparing A-DLISTA with its non-augmented version, DLISTA, one can notice the benefits of using an augmentation network to make the model adaptive. Concerning Bayesian approaches, VLISTA outperforms BCS. However, it is important to note that BC does not have trainable parameters, unlike VLISTA. Therefore, the higher performance of VLISTA comes at the price of an expensive training procedure. Similar to the synthetic dataset, VLISTA exhibits a drop in performance compared to A-DLISTA for MNIST and CIFAR10.

**Table 1: MNIST SSIM (the higher the better) for different number of measurements**. First three rows correspond to "classical" baselines. We highlight in bold the best performance for Bayes and Non-Bayes models.

| | | SSIM ↑ | | | | |
|---|---|---|---|---|---|---|
| | | number of measurements | | | | |
| | | 1 $(\times e^{-1})$ | 10 $(\times e^{-1})$ | 100 $(\times e^{-1})$ | 300 $(\times e^{-1})$ | 500 $(\times e^{-1})$ |
| Non-Bayes | Canonical | $0.39_{\pm 0.12}$ | $0.56_{\pm 0.04}$ | $2.20_{\pm 0.04}$ | $3.75_{\pm 0.05}$ | $4.94_{\pm 0.06}$ |
| | Wavelet | $0.40_{\pm 0.09}$ | $0.56_{\pm 0.06}$ | $2.30_{\pm 0.06}$ | $3.90_{\pm 0.05}$ | $5.05_{\pm 0.01}$ |
| | SPCA | $0.45_{\pm 0.11}$ | $0.65_{\pm 0.06}$ | $2.72_{\pm 0.06}$ | $3.52_{\pm 0.08}$ | $4.98_{\pm 0.08}$ |
| | LISTA | $\mathbf{0.96_{\pm 0.01}}$ | $1.11_{\pm 0.01}$ | $3.70_{\pm 0.01}$ | $5.36_{\pm 0.01}$ | $6.31_{\pm 0.01}$ |
| | DLISTA | $\mathbf{0.96_{\pm 0.01}}$ | $1.09_{\pm 0.01}$ | $4.01_{\pm 0.02}$ | $5.57_{\pm 0.01}$ | $6.26_{\pm 0.01}$ |
| | A-DLISTA (our) | $\mathbf{0.96_{\pm 0.01}}$ | $\mathbf{1.17_{\pm 0.01}}$ | $\mathbf{4.79_{\pm 0.01}}$ | $\mathbf{6.15_{\pm 0.01}}$ | $\mathbf{6.70_{\pm 0.01}}$ |
| Bayes | BCS | $0.05_{\pm 0.01}$ | $0.60_{\pm 0.01}$ | $1.10_{\pm 0.01}$ | $4.48_{\pm 0.02}$ | $\mathbf{6.23_{\pm 0.02}}$ |
| | VLISTA (our) | $\mathbf{0.80_{\pm 0.03}}$ | $\mathbf{0.94_{\pm 0.02}}$ | $\mathbf{3.29_{\pm 0.01}}$ | $\mathbf{4.73_{\pm 0.01}}$ | $6.02_{\pm 0.01}$ |

## 5.4 Out Of Distribution Detection

This section focuses on a crucial distinction between non-Bayesian models and VLISTA for solving inverse linear problems. Unlike any non-Bayesian approach to compressed sensing, VLISTA allows quantifying uncertainties on the reconstructed signals. This means that it can detect out-of-distribution samples without requiring ground truth data during inference. In contrast to other Bayesian techniques that design specific priors to meet the sparsity constraints after marginalization (Ji et al., 2008; Zhou et al., 2014), VLISTA completely overcomes such an issue as the thresholding operations are not affected by the marginalization over dictionaries. To prove that VLISTA can detect OOD samples, we employ the MNIST dataset. First, we split the dataset into two distinct subsets - the In-Distribution (ID) set and the OOD. The ID set comprises images from three randomly chosen digits, while the OOD set includes images of the remaining digits. Then, we partitioned the ID set into training, validation, and test sets for VLISTA. Once the model was trained,

**Table 2: CIFAR10 SSIM (the higher the better) for different number of measurements**. First three rows correspond to "classical" baselines. We highlight in bold the best performance for Bayes and Non-Bayes models.

| | | SSIM ↑ | | | | |
|---|---|---|---|---|---|---|
| | | number of measurements | | | | |
| | | 1 $(\times e^{-1})$ | 10 $(\times e^{-1})$ | 100 $(\times e^{-1})$ | 300 $(\times e^{-1})$ | 500 $(\times e^{-1})$ |
| Non-Bayes | Canonical | $0.17_{\pm 0.10}$ | $0.21_{\pm 0.02}$ | $0.33_{\pm 0.02}$ | $0.47_{\pm 0.02}$ | $0.58_{\pm 0.03}$ |
| | Wavelet | $0.23_{\pm 0.22}$ | $0.42_{\pm 0.02}$ | $1.44_{\pm 0.06}$ | $2.52_{\pm 0.09}$ | $3.43_{\pm 0.08}$ |
| | SPCA | $0.31_{\pm 0.19}$ | $0.43_{\pm 0.02}$ | $1.53_{\pm 0.04}$ | $2.66_{\pm 0.08}$ | $3.58_{\pm 0.07}$ |
| | LISTA | $\mathbf{1.34_{\pm 0.02}}$ | $1.67_{\pm 0.02}$ | $3.10_{\pm 0.01}$ | $4.20_{\pm 0.01}$ | $4.71_{\pm 0.01}$ |
| | DLISTA | $1.16_{\pm 0.02}$ | $1.96_{\pm 0.02}$ | $4.50_{\pm 0.01}$ | $5.15_{\pm 0.01}$ | $5.42_{\pm 0.01}$ |
| | A-DLISTA (our) | $\mathbf{1.34_{\pm 0.02}}$ | $\mathbf{1.77_{\pm 0.02}}$ | $\mathbf{4.74_{\pm 0.01}}$ | $\mathbf{5.26_{\pm 0.01}}$ | $\mathbf{5.83_{\pm 0.01}}$ |
| Bayes | BCS | $0.04_{\pm 0.01}$ | $0.48_{\pm 0.01}$ | $0.59_{\pm 0.01}$ | $1.29_{\pm 0.01}$ | $1.91_{\pm 0.01}$ |
| | VLISTA (our) | $\mathbf{0.86_{\pm 0.03}}$ | $\mathbf{1.25_{\pm 0.03}}$ | $\mathbf{3.59_{\pm 0.02}}$ | $\mathbf{4.01_{\pm 0.01}}$ | $\mathbf{4.36_{\pm 0.01}}$ |

it was tasked with reconstructing images from the ID test and OOD sets. To assess the model's ability to detect OODs, we utilized a *two-sample t-test*. We accomplished that by leveraging the per-pixel variance of the reconstructed ID, $\{var_{\sigma_{pp}}^{ID_{test};i}\}_{i=0}^{P-1}$, and OOD, $\{var_{\sigma_{pp}}^{OOD;i}\}_{i=0}^{P-1}$, images (with $P$ being the number of pixels). To compute the per-pixel variance, we reconstruct each image 100 times by sampling a different dictionary for each of trial. We then construct the empirical c.d.f. of the per-pixel variance for each image. By using the mean of the c.d.f. as a summary statistics, we can apply the *two-sample t-test* to detect OOD samples. We report the results in Figure 4. As a reference *p*-value for rejecting the null hypothesis about the two variance distributions being the same, we consider a significance level equal to 0.05 (green solid line). We conducted multiple tests at different noise levels to assess the robustness of OOD detection to measure noise. For the current task, we used BCS as a baseline. However, due to the different nature of the BCS framework, we utilized a slightly different evaluation procedure to determine its *p*-values. We employed the same ID and OOD splits as VLISTA but considered the c.d.f. of the reconstruction error the model evaluates. The remainder of the process was identical to that of VLISTA.

## 6 Conclusion

Our study introduces a novel approach called VLISTA, which combines dictionary learning and sparse recovery into a single variational framework. Traditional compressed sensing methods rely on a known ground truth dictionary to reconstruct signals. Moreover state-of-the-art LISTA-type of models, typically assume a fixed measurement setup. In our work, we relax both assumptions. First, we propose a soft-thresholding algorithm, termed A-DLISTA, that can handle different sensing matrices. We theoretically justify the use of an augmentation network to adapt the threshold and step size for each layer based on the current input and the learned dictionary. Finally, we propose a probabilistic assumption about the existence of a ground truth dictionary and use it to create

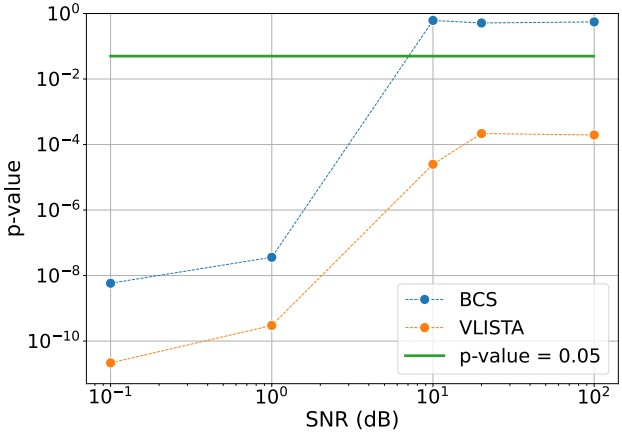

**Figure 4: p-value for OOD rejection as a function of the noise level**. The green line represents a reference p-value equal to 0.05.

the VLISTA framework. Our empirical results show that A-DLISTA improves upon performances of classical and ML baselines in a non-static measurement scenario. Although VLISTA does not outperform A-DLISTA, it allows for uncertainty evaluation in reconstructed signals, a valuable feature for detecting out-of-distribution data. In contrast, none of the non-Bayesian models can perform such a task. Unlike other Bayesian approaches, VLISTA does not require specific priors to preserve sparsity after marginalization. Instead, the averaging operation applies to the sparsifying dictionary, not the sparse signal itself.

## Impact Statement

This work proposes two new models to jointly solve the dictionary learning and sparse recovery problems, especially concerning scenarios characterized by a varying sensing matrix. We believe the potential societal consequences of our work being chiefly positive, since it might contribute to a larger adoption of LISTA-type of models to applications requiring fast solutions to underdetermined inverse problems, especially concerning varying forward operators. Nonetheless, it is crucial to exercise caution and thoroughly comprehend the behavior of A-DLISTA and VLISTA, as with any other LISTA model, in order to obtain reliable predictions.

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

## A    Appendix

## B    Theoretical Analysis

In this section, we report about theoretical motivations for the A-DLISTA design choices. The design is motivated by considering the convergence analysis of LISTA method. We start by recalling a result from Chen et al. (2018), upon which our analysis relies. The authors of Chen et al. (2018) consider the inverse problem $\boldsymbol{y} = \boldsymbol{A}\boldsymbol{x}_*$, with $\boldsymbol{x}_*$ as the ground truth sparse vector, and use the model with each layer given by:

$$\boldsymbol{x}_t = \eta_{\theta_t}\left(\boldsymbol{x}_{t-1} + \boldsymbol{W}_t^\top(\boldsymbol{y} - \boldsymbol{A}\boldsymbol{x}_{t-1})\right), \tag{16}$$

where $(\boldsymbol{W}_t, \theta_t)$ are learnable parameters.

The following result from Chen et al. (2018, Theorem 2) is adapted to noiseless setting.

**Theorem B.1.** *Suppose that the iterations of LISTA are given by equation 16, and assume $\|\boldsymbol{x}_*\|_\infty \leq B$ and $\|\boldsymbol{x}_*\|_0 \leq s$. There exists a sequence of parameters $\{\boldsymbol{W}_t, \theta_t\}$ such that*

$$\|\boldsymbol{x}_t - \boldsymbol{x}_*\|_2 \leq sB\exp(-ct), \quad \forall t = 1, 2, \ldots,$$

*for constant $c > 0$ that depend only on the sensing matrix $\boldsymbol{A}$ and the sparsity level $s$.*

It is important to note that the above convergence result only assures the *existence* of the parameters that are good for convergence but does not guarantee that the training would necessarily find them. The latter result is in general difficult to obtain.

The proof in Chen et al. (2018) has two main steps:

1. No false positive: the thresholds are chosen such that the entries outside the support of $\boldsymbol{x}_*$ remain zero. The choice of threshold, among others, depend on the coherence value of $\boldsymbol{W}_t$ and $\boldsymbol{A}$. We will provide more details below.

2. Error bounds for $\boldsymbol{x}_*$: assuming proper choice of thresholds, the authors derive bounds on the recovery error.

We focus on adapting these steps to our setup. Note that to assure there is no false positive, it is common in classical ISTA literature to start from large thresholds, so the soft thresholding function aggressively maps many entries to zero, and then gradually reduce the threshold value as the iterations progress.

### B.1    Analysis with known ground-truth dictionary

Let's consider the extension of Theorem B.1 to our setup:

$$\boldsymbol{x}_t = \eta_{\theta_t}\left(\boldsymbol{x}_{t-1} + \gamma_t(\boldsymbol{\Phi}^k\boldsymbol{\Psi}_t)^T(\boldsymbol{y}^k - \boldsymbol{\Phi}^k\boldsymbol{\Psi}_t\boldsymbol{x}_{t-1})\right). \tag{17}$$

Note that in our case, the weight $\boldsymbol{W}_t$ is replaced with $\gamma_t(\boldsymbol{\Phi}^k\boldsymbol{\Psi}_t)$ with learnable $\boldsymbol{\Psi}_t$ and $\gamma_t$. Besides, the matrix $\boldsymbol{A}$ is replaced by $\boldsymbol{\Phi}^k\boldsymbol{\Psi}_t$, and the forward model is given by $\boldsymbol{y}^k = \boldsymbol{\Phi}^k\boldsymbol{\Psi}_o\boldsymbol{x}_*$. The sensing matrix $\boldsymbol{\Phi}^k$ can change across samples, hence the dependence on the sample index $k$.

If the learned dictionary $\boldsymbol{\Psi}_t$ is equal to $\boldsymbol{\Psi}_o$, the layers of our model are equal to classical iterative soft-thresholding algorithms with learnable step-size $\gamma_t$ and threshold $\theta_t$.

There are many convergence results in the literature, for example see Daubechies et al. (2004). We can use convergence analysis of iterative soft thresholding algorithms using the mutual coherence similar to Chen et al. (2018); Behrens et al. (2021). As a reminder, the mutual coherence of the matrix $\boldsymbol{M}$ is defined as:

$$\mu(\boldsymbol{M}) := \max_{1 \leq i \neq j \leq N}\left|\boldsymbol{M}_i^\top\boldsymbol{M}_j\right|, \tag{18}$$

where $\boldsymbol{M}_i$ is the $i$'th column of $\boldsymbol{M}$.

The convergence result requires that the mutual coherence $\mu(\boldsymbol{\Phi}^k \boldsymbol{\Psi}_o)$ be sufficiently small, for example in order of $1/(2s)$ with $s$ the sparsity, and the matrix $\boldsymbol{\Phi}^k \boldsymbol{\Psi}_o$ is column normalized, i.e., $\left\| (\boldsymbol{\Phi}^k \boldsymbol{\Psi}_o) \right\|_2 = 1$. Then the step size can be chosen equal to one, i.e., $\gamma_t = 1$. The thresholds $\theta_t$ are chosen to avoid false positive using a similar schedule mentioned above, that is, first starting with a large threshold $\theta_0$ and then gradually decreasing it to a certain limit. We do not repeat the derivations, and interested readers can refer to Daubechies et al. (2004); Behrens et al. (2021); Chen et al. (2018) and references therein.

*Remark* B.2. When the dictionary $\boldsymbol{\Psi}_o$ is known, we can adapt the algorithm to the varying sensing matrix $\boldsymbol{\Phi}^k$ by first normalizing the column $\boldsymbol{\Phi}^k \boldsymbol{\Psi}_o$. What is important to note is that the threshold choice is a function the mutual coherence of the sensing matrix. So with each new sensing matrix, the thresholds should be adapted following the mutual coherence value. This observation partially justifies the choice of thresholds as a function of the dictionary and the sensing matrix, hence the augmentation network.

## B.2 Analysis with unknown dictionary

We now move to the scenario where the dictionary is itself learned, and not known in advance.

Consider the layer $t$ of DLISTA with the sensing matrix $\boldsymbol{\Phi}^k$, and define the following parameters:

$$\tilde{\mu}(t, \boldsymbol{\Phi}^k) := \max_{1 \le i \ne j \le N} \left| ((\boldsymbol{\Phi}^k \boldsymbol{\Psi}_t)_i)^\top (\boldsymbol{\Phi}^k \boldsymbol{\Psi}_t)_j \right| \tag{19}$$

$$\tilde{\mu}_2(t, \boldsymbol{\Phi}^k) := \max_{1 \le i,j \le N} \left| ((\boldsymbol{\Phi}^k \boldsymbol{\Psi}_t)_i)^\top (\boldsymbol{\Phi}^k (\boldsymbol{\Psi}_t - \boldsymbol{\Psi}_o))_j \right| \tag{20}$$

$$\delta(\gamma, t, \boldsymbol{\Phi}^k) := \max_i \left| 1 - \gamma \left\| (\boldsymbol{\Phi}^k \boldsymbol{\Psi}_t)_i \right\|_2^2 \right| \tag{21}$$

Some comments are in order:

- The term $\tilde{\mu}$ is the **mutual coherence** of the matrix $\boldsymbol{\Phi}^k \boldsymbol{\Psi}_t$.

- The term $\tilde{\mu}_2$ is closely connected to **generalized mutual coherence**, however, it differs in that unlike generalized mutual coherence, it includes the diagonal inner product for $i = j$. It captures the effect of mismatch with ground-truth dictionary.

- Finally, the term $\delta(\cdot)$ is reminiscent of restricted isometry property (RIP) constant (Foucart & Rauhut, 2013), a key condition for many recovery guarantees in compressed sensing. When the columns of the matrix $\boldsymbol{\Phi}^k \boldsymbol{\Psi}_t$ is normalized, the choice of $\gamma = 1$ yield $\delta(\gamma, t, \boldsymbol{\Phi}^k) = 0$.

For the rest of the paper, for simplicity, we only kept the dependence on $\gamma$ in the notation and dropped the dependence of $\tilde{\mu}, \tilde{\mu}_2$ and $\delta$ on $t$, $\boldsymbol{\Phi}^k$ and $\boldsymbol{\Psi}_t$.

**Proposition B.3.** *Suppose that $\boldsymbol{y}^k = \boldsymbol{\Phi}^k \boldsymbol{\Psi}_o \boldsymbol{x}_*$ with the support $supp(\boldsymbol{x}_*) = S$. For DLISTA iterations give as*

$$\boldsymbol{x}_t = \eta_{\theta_t} \left( \boldsymbol{x}_{t-1} + \gamma_t (\boldsymbol{\Phi}^k \boldsymbol{\Psi}_t)^T (\boldsymbol{y}^k - \boldsymbol{\Phi}^k \boldsymbol{\Psi}_t \boldsymbol{x}_{t-1}) \right), \tag{22}$$

*we have:*

1. *If for all $t$, the pairs $(\theta_t, \gamma_t, \boldsymbol{\Psi}_t)$ satisfy*

$$\gamma_t \left( \tilde{\mu} \left\| \boldsymbol{x}_* - \boldsymbol{x}_{t-1} \right\|_1 + \tilde{\mu}_2 \left\| \boldsymbol{x}_* \right\|_1 \right) \le \theta_t, \tag{23}$$

   *then there is no false positive in each iteration. In other words, for all $t$, we have $supp(\boldsymbol{x}_t) \subseteq supp(\boldsymbol{x}_*)$.*

2. *Assuming that the conditions of the last step hold, then we get the following bound on the error:*

$$\left\| \boldsymbol{x}_t - \boldsymbol{x}_* \right\|_1 \le (\delta(\gamma_t) + \gamma_t \tilde{\mu}(|S| - 1)) \left\| \boldsymbol{x}_{t-1} - \boldsymbol{x}_* \right\|_1 + \gamma_t \tilde{\mu}_2 |S| \left\| \boldsymbol{x}_* \right\|_1 + |S| \theta_t.$$

### B.2.1 Guidelines from Proposition.

We remark on some of the guidelines we can get from the above result.

- **Thresholds.** Similar to the discussion in previous sections, there are thresholds such that, there is no false positive at each layer. The choice of $\theta_t$ is a function of $\gamma_t$ and, through coherence terms, $\boldsymbol{\Phi}^k$ and $\boldsymbol{\Psi}_t$. Since $\boldsymbol{\Phi}^k$ changes for each sample $k$, we learn a neural network that yields this parameter as a function of $\boldsymbol{\Phi}^k$ and $\boldsymbol{\Psi}_t$.

- **Step size.** The step size $\gamma_t$ can be chosen to control the error decay. Ideally, we would like to have the term $(\delta(\gamma_t) + \gamma_t\tilde{\mu}(|S| - 1))$ to be strictly smaller than one. In particular, $\gamma_t$ directly impacts $\delta(\gamma_t)$, also a function of $\boldsymbol{\Phi}^k$ and $\boldsymbol{\Psi}_t$. We can therefore consider $\gamma_t$ as a function of $\boldsymbol{\Phi}^k$ and $\boldsymbol{\Psi}_t$, which hints at the augmentation neural network we introduced for giving $\gamma_t$ as a function of those parameters.

**Remarks on Convergence.** One might wonder if the convergence is possible given the bound on the error. We try to sketch a scenario where this can happen. First, note that once we have chosen $\gamma_t$, and given $\Phi_k$ and $\Psi_t$, we can select $\theta_t$ using the condition 23. Also, if the network gradually learns the ground truth dictionary at later stages, the term $\tilde{\mu}_2$ vanishes. We need to choose the term $\gamma_t$ carefully such that the term $(\delta(\gamma_t) + \gamma_t\tilde{\mu}(|S| - 1))$ is smaller than one. Similar to ISTA analysis, we would need to assume bounds on the mutual coherence $\tilde{\mu}$ and the column norm for $\boldsymbol{\Phi}^k\boldsymbol{\Psi}_o$. With standard assumptions, sketched above as well, the error gradually decreases per iteration, and we can reuse the convergence results of ISTA. We would like to emphasize that this is a heuristic argument, and there is no guarantee that the training yields a model with the parameters in accordance with these guidelines. Although we show experimentally that the proposed methods provide the promised improvements.

### B.3 Proof of Proposition B.3

In what follows, we provide the derivations for Proposition B.3.

Convergence proofs of ISTA type models involve two steps in general. First, it is investigated how the support is found and locked in, and second how the error shrinks at each step. We focus on these two steps, which matter mainly for our architecture design. Our analysis is similar in nature to Chen et al. (2018); Aberdam et al. (2021), however it differs from Aberdam et al. (2021) in considering unknown dictionaries and from Chen et al. (2018) in both considered architecture and varying sensing matrix. In what follows, we consider noiseless setting. However, the results can be extended to noisy setups by adding additional terms containing noise norm similar to Chen et al. (2018). We make following assumptions:

1. There is a ground-truth (unknown) dictionary $\boldsymbol{\Psi}_o$ such that $\boldsymbol{s}_* = \boldsymbol{\Psi}_o\boldsymbol{x}_*$.

2. As a consequence, $\boldsymbol{y}^k = \boldsymbol{\Phi}^k\boldsymbol{\Psi}_o\boldsymbol{x}_*$.

3. We assume that $\boldsymbol{x}_*$ is sparse with its support contained in $S$. In other words: $x_{i,*} = 0$ for $i \notin S$.

To simplify the notation, we drop the index $k$, which indicates the varying sensing matrix, from $\boldsymbol{\Phi}^k$ and $\boldsymbol{y}^k$, and use $\boldsymbol{\Phi}$ and $\boldsymbol{y}$ for the rest. We break the proof to two lemma, each proving one part of Proposition B.3.

### B.3.1 Proof - step 1: no false positive condition

The following lemma focuses on assuring that we do not have false positive in support recovery after each iteration of our model. In other words, the models continues updating only the entries in the support and keep the entries outside the support zero.

**Lemma B.4.** *Suppose that the support of $\boldsymbol{x}_*$ is given as $supp(\boldsymbol{x}_*) = S$. Consider iterations given by:*

$$\boldsymbol{x}_t = \eta_{\theta_t}\left(\boldsymbol{x}_{t-1} + \gamma_t(\boldsymbol{\Phi}\boldsymbol{\Psi}_t)^\top(\boldsymbol{y} - \boldsymbol{\Phi}\boldsymbol{\Psi}_t\boldsymbol{x}_{t-1})\right),$$

*with $\boldsymbol{x}_0 = 0$. If we have for all $t = 1, 2, \ldots$:*

$$\gamma_t \left( \tilde{\mu} \|\boldsymbol{x}_* - \boldsymbol{x}_{t-1}\|_1 + \tilde{\mu}_2 \|\boldsymbol{x}_*\|_1 \right) \leq \theta_t,$$

*then there will be no false positive, i.e., $x_{t,i} = 0$ for $\forall i \notin S, \forall t$.*

*Proof.* We prove this by induction. Since $\boldsymbol{x}_0 = 0$, the induction base is trivial. Suppose that the support of $\boldsymbol{x}_{t-1}$ is already included in that of $\boldsymbol{x}_*$, namely $\mathrm{supp}(\boldsymbol{x}_{t-1}) \subseteq \mathrm{supp}(\boldsymbol{x}_*) = S$. Consider $i \in S^c$. We have

$$x_{t,i} = \eta_{\theta_t} \left( \gamma_t ((\boldsymbol{\Phi}\boldsymbol{\Psi}_t)_i)^\top (\boldsymbol{y} - \boldsymbol{\Phi}\boldsymbol{\Psi}_t \boldsymbol{x}_{t-1}) \right). \tag{24}$$

To avoid false positives, we need to guarantee that for $i \notin S$:

$$\eta_{\theta_t} \left( \gamma_t ((\boldsymbol{\Phi}\boldsymbol{\Psi}_t)_i)^\top (\boldsymbol{y} - \boldsymbol{\Phi}\boldsymbol{\Psi}_t \boldsymbol{x}_{t-1}) \right) = 0 \implies \left| \gamma_t ((\boldsymbol{\Phi}\boldsymbol{\Psi}_t)_i)^\top (\boldsymbol{y} - \boldsymbol{\Phi}\boldsymbol{\Psi}_t \boldsymbol{x}_{t-1}) \right| \leq \theta_t, \tag{25}$$

which means that the soft-thresholding function will have zero output for these entries. First note that:

$$\left| ((\boldsymbol{\Phi}\boldsymbol{\Psi}_t)_i)^\top \boldsymbol{\Phi}(\boldsymbol{\Psi}_o \boldsymbol{x}_* - \boldsymbol{\Psi}_t \boldsymbol{x}_{t-1}) \right| \leq \left| ((\boldsymbol{\Phi}\boldsymbol{\Psi}_t)_i)^\top \boldsymbol{\Phi}(\boldsymbol{\Psi}_t \boldsymbol{x}_* - \boldsymbol{\Psi}_t \boldsymbol{x}_{t-1}) \right|$$
$$+ \left| ((\boldsymbol{\Phi}\boldsymbol{\Psi}_t)_i)^\top \boldsymbol{\Phi}(\boldsymbol{\Psi}_o \boldsymbol{x}_* - \boldsymbol{\Psi}_t \boldsymbol{x}_*) \right| \tag{26}$$

$$= \left| \sum_{j \in S} ((\boldsymbol{\Phi}\boldsymbol{\Psi}_t)_i)^\top (\boldsymbol{\Phi}\boldsymbol{\Psi}_t)_j (\boldsymbol{x}_{*,j} - \boldsymbol{x}_{t-1,j}) \right| + \left| ((\boldsymbol{\Phi}\boldsymbol{\Psi}_t)_i)^\top \boldsymbol{\Phi}(\boldsymbol{\Psi}_o \boldsymbol{x}_* - \boldsymbol{\Psi}_t \boldsymbol{x}_*) \right| \tag{27}$$

We can bound the first term by:

$$\left| \sum_{j \in S} ((\boldsymbol{\Phi}\boldsymbol{\Psi}_t)_i)^\top (\boldsymbol{\Phi}\boldsymbol{\Psi}_t)_j (\boldsymbol{x}_{*,j} - \boldsymbol{x}_{t-1,j}) \right| \leq \sum_{j \in S} \left| ((\boldsymbol{\Phi}\boldsymbol{\Psi}_t)_i)^\top (\boldsymbol{\Phi}\boldsymbol{\Psi}_t)_j \right| \left| (\boldsymbol{x}_{*,j} - \boldsymbol{x}_{t-1,j}) \right|$$
$$\leq \tilde{\mu} \|\boldsymbol{x}_* - \boldsymbol{x}_{t-1}\|_1,$$

where we use the definition of mutual coherence for the upper bound. The last term is bounded by

$$\left| ((\boldsymbol{\Phi}\boldsymbol{\Psi}_t)_i)^\top \boldsymbol{\Phi}(\boldsymbol{\Psi}_o \boldsymbol{x}_* - \boldsymbol{\Psi}_t \boldsymbol{x}_*) \right| = \left| \sum_{j \in S} ((\boldsymbol{\Phi}\boldsymbol{\Psi}_t)_i)^\top (\boldsymbol{\Phi}(\boldsymbol{\Psi}_o - \boldsymbol{\Psi}_t))_j x_{j,*} \right| \tag{28}$$

$$\leq \sum_{j \in S} \left| ((\boldsymbol{\Phi}\boldsymbol{\Psi}_t)_i)^\top (\boldsymbol{\Phi}(\boldsymbol{\Psi}_o - \boldsymbol{\Psi}_t))_j \right| |x_{j,*}| \tag{29}$$

$$\leq \tilde{\mu}_2 \|\boldsymbol{x}_*\|_1. \tag{30}$$

Therefore, we get

$$\left| \gamma_t ((\boldsymbol{\Phi}\boldsymbol{\Psi}_t)_i)^\top (\boldsymbol{y} - \boldsymbol{\Phi}\boldsymbol{\Psi}_t \boldsymbol{x}_{t-1}) \right| \leq \gamma_t \left( \tilde{\mu} \|\boldsymbol{x}_* - \boldsymbol{x}_{t-1}\|_1 + \tilde{\mu}_2 \|\boldsymbol{x}_*\|_1 \right)$$

The following choice guarantees that there is no false positive:

$$\gamma_t \left( \tilde{\mu} \|\boldsymbol{x}_* - \boldsymbol{x}_{t-1}\|_1 + \tilde{\mu}_2 \|\boldsymbol{x}_*\|_1 \right) \leq \theta_t. \tag{31}$$

$\square$

### B.3.2  Proof - step 2: controlling the recovery error

The previous lemma provided the conditions such that there is no false positive. We see under which conditions the model can reduce the error inside the support $S$.

**Lemma B.5.** *Suppose that the threshold parameter $\theta_t$ has been chosen such that there is no false positive after each iteration. We have:*

$$\|\boldsymbol{x}_t - \boldsymbol{x}_*\|_1 \leq (\delta(\gamma_t) + \gamma_t \tilde{\mu}(|S| - 1)) \|\boldsymbol{x}_{t-1} - \boldsymbol{x}_*\|_1 + \gamma_t \tilde{\mu}_2 |S| \|\boldsymbol{x}_*\|_1 + |S|\theta_t.$$

*Proof.* For $i \in S$, we have:

$$|x_{t,i} - x_{*,i}| \leq |x_{t-1,i} + \gamma_t((\boldsymbol{\Phi}\boldsymbol{\Psi}_t)_i)^\top (\boldsymbol{y} - \boldsymbol{\Phi}\boldsymbol{\Psi}_t \boldsymbol{x}_{t-1}) - x_{*,i}| + \theta_t. \tag{32}$$

At the iteration $t$ for $i \in S$, we can separate the dictionary mismatch and the rest of the error as follows:

$$x_{t-1,i} + \gamma_t((\boldsymbol{\Phi}\boldsymbol{\Psi}_t)_i)^\top (\boldsymbol{y} - \boldsymbol{\Phi}\boldsymbol{\Psi}_t \boldsymbol{x}_{t-1}) =$$

$$x_{t-1,i} + \gamma_t(\sum_{j \in S}((\boldsymbol{\Phi}\boldsymbol{\Psi}_t)_i)^\top (\boldsymbol{\Phi}\boldsymbol{\Psi}_t)_j(\boldsymbol{x}_{*,j} - \boldsymbol{x}_{t-1,j}) + ((\boldsymbol{\Phi}\boldsymbol{\Psi}_t)_i)^\top \boldsymbol{\Phi}(\boldsymbol{\Psi}_o \boldsymbol{x}_* - \boldsymbol{\Psi}_t \boldsymbol{x}_*)).$$

We can decompose the first part further as:

$$x_{t-1,i} + \gamma_t \sum_{j \in S}((\boldsymbol{\Phi}\boldsymbol{\Psi}_t)_i)^\top (\boldsymbol{\Phi}\boldsymbol{\Psi}_t)_j(x_{*,j} - \boldsymbol{x}_{t-1,j}) =$$

$$(\boldsymbol{I} - \gamma_t(\boldsymbol{\Phi}\boldsymbol{\Psi}_t)_i)^\top (\boldsymbol{\Phi}\boldsymbol{\Psi}_t)_i))\boldsymbol{x}_{t-1,i} + \gamma_t(\boldsymbol{\Phi}\boldsymbol{\Psi}_t)_i)^\top (\boldsymbol{\Phi}\boldsymbol{\Psi}_t)_i)\boldsymbol{x}_{*,i}$$

$$+ \gamma_t \sum_{j \in S, j \neq i}((\boldsymbol{\Phi}\boldsymbol{\Psi}_t)_i)^\top (\boldsymbol{\Phi}\boldsymbol{\Psi}_t)_j(\boldsymbol{x}_{*,j} - \boldsymbol{x}_{t-1,j}).$$

Using triangle inequality for the previous decomposition we get:

$$|x_{t-1,i} + \gamma_t((\boldsymbol{\Phi}\boldsymbol{\Psi}_t)_i)^\top (\boldsymbol{y} - \boldsymbol{\Phi}\boldsymbol{\Psi}_t \boldsymbol{x}_{t-1}) - x_{*,i}| \leq |(1 - \gamma_t(\boldsymbol{\Phi}\boldsymbol{\Psi}_t)_i)^\top (\boldsymbol{\Phi}\boldsymbol{\Psi}_t)_i))(x_{t-1,i} - x_{*,i})|$$

$$+ \gamma_t \left| \sum_{j \in S, j \neq i}((\boldsymbol{\Phi}\boldsymbol{\Psi}_t)_i)^\top (\boldsymbol{\Phi}\boldsymbol{\Psi}_t)_j(\boldsymbol{x}_{*,j} - \boldsymbol{x}_{t-1,j}) \right|$$

$$+ \gamma_t \left| ((\boldsymbol{\Phi}\boldsymbol{\Psi}_t)_i)^\top \boldsymbol{\Phi}(\boldsymbol{\Psi}_o \boldsymbol{x}_* - \boldsymbol{\Psi}_t \boldsymbol{x}_*)) \right|$$

$$\leq \delta(\gamma_t) |(z_{t-1,i} - z_{*,i})|$$

$$+ \gamma_t \sum_{j \in S, j \neq i} \tilde{\mu} |x_{*,j} - x_{t-1,j}| + \gamma_t \tilde{\mu}_2 \|\boldsymbol{x}_*\|_1$$

It suffices to sum up the errors and combine previous inequalities to get:

$$\|\boldsymbol{x}_{S,t} - \boldsymbol{x}_*\|_1 = \sum_{i \in S} |\boldsymbol{x}_{t,i} - \boldsymbol{x}_{*,i}| \leq$$

$$\leq (\delta(\gamma_t) + \gamma_t \tilde{\mu}(|S| - 1)) \|\boldsymbol{x}_{S,t-1} - \boldsymbol{x}_*\|_1 + \gamma_t \tilde{\mu}_2 |S| \|\boldsymbol{x}_*\|_1 + |S|\theta_t.$$

Since we assumed there is no false positive, we get the final result:

$$\|\boldsymbol{x}_t - \boldsymbol{x}_*\|_1 = \sum_{i \in S} |\boldsymbol{x}_{t,i} - \boldsymbol{x}_{*,i}| \leq (\delta(\gamma_t) + \gamma_t \tilde{\mu}(|S| - 1)) \|\boldsymbol{x}_{t-1} - \boldsymbol{x}_*\|_1 + \gamma_t \tilde{\mu}_2 |S| \|\boldsymbol{x}_*\|_1 + |S|\theta_t.$$

$\square$

# C Implementation Details

In this section we report details concerning the architecture of A-DLISTA and VLISTA.

## C.1 A-DLISTA (Augmentation Network)

As previously stated in the main paper (subsection 4.1), A-DLISTA consists of two architectures: the DLISTA model (blue blocks in Figure 1) representing the unfolded version of the ISTA algorithm with parametrized $\boldsymbol{\Psi}$, and the augmentation (or adaptation) network (red blocks in Figure 1). At a given reconstruction layer $t$, the augmentation model takes the measurement matrix $\boldsymbol{\Phi}^i$ and the dictionary $\boldsymbol{\Psi}_t$ as input and generates the parameters $\{\gamma_t, \theta_t\}$ for the current iteration. The architecture for the augmentation network is illustrated in Figure 5, which shows a feature extraction section and two output branches, one for each generated parameter. To ensure that the estimated $\{\gamma_t, \theta_t\}$ parameters are positive, each branch is equipped with a softplus function. As noted in the main paper, the weights of the augmentation model are shared across all A-DLISTA layers.

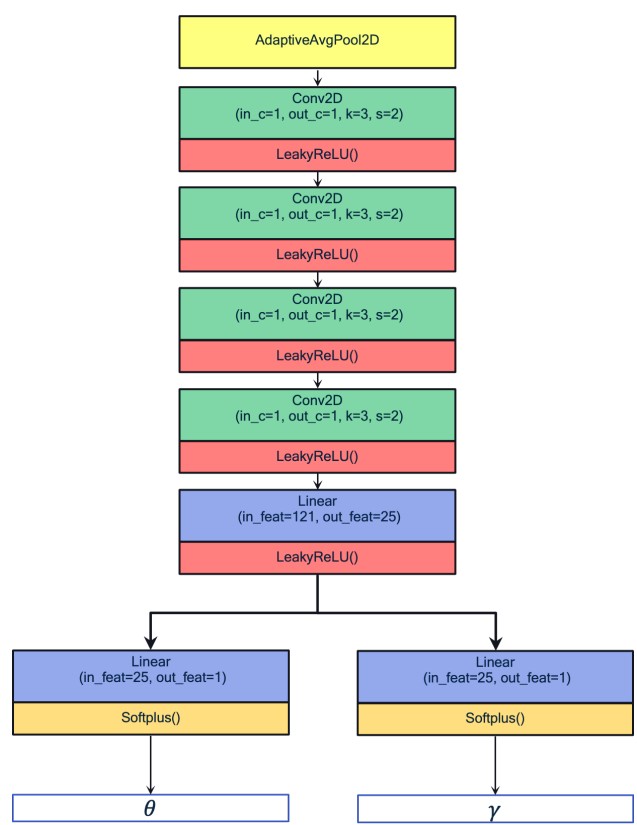

**Figure 5:** Augmentation model's architecture for A-DLISTA.

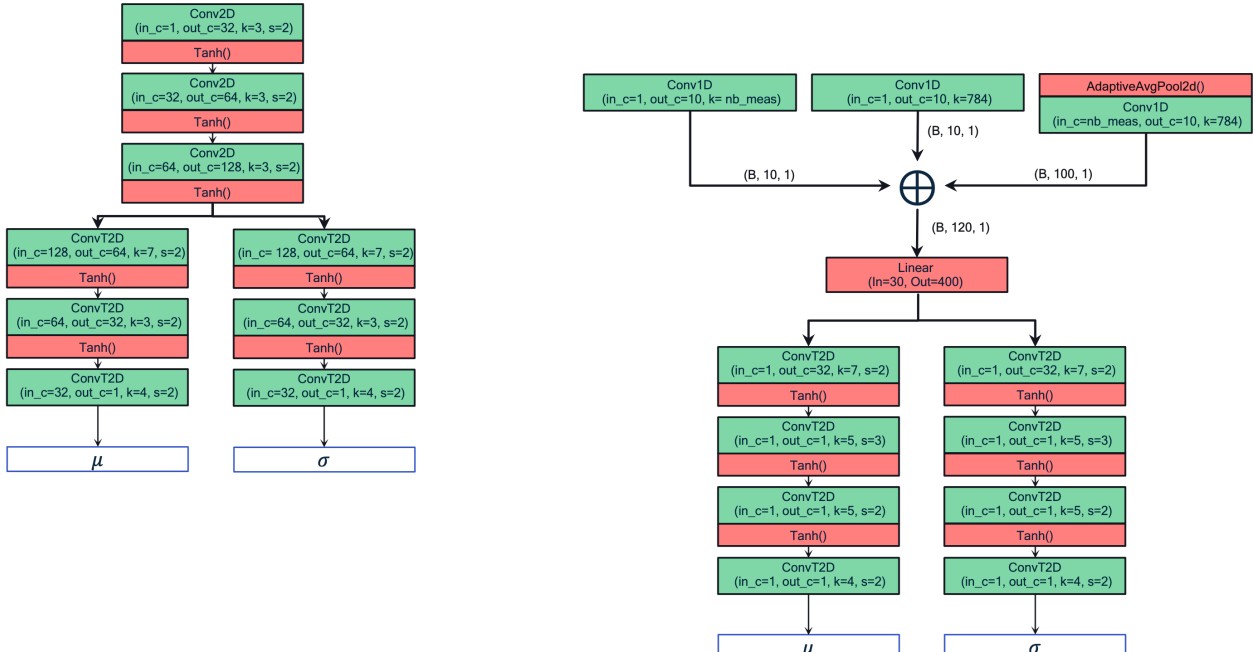

**Figure 6:** Left: prior network architecture. Right: posterior network architecture. For the posterior model, we show the output shape from each of the three input heads in the figure. Such a structure is necessary since the posterior model accepts three quantities as input: observations, the sensing matrix, and the reconstruction from the previous layer. Different shapes characterize these quantities. The letter "B" indicates batch size.

## C.2  VLISTA

As described in subsection 4.2 of the meain paper, VLISTA comprises three different components: the likelihood, and the prior and posterior models.

### C.2.1  VLISTA - Likelihood model

The likelihood model (subsection 4.2) represents a Gaussian distribution with a mean value parametrized using the A-DLISTA model. There is. However, a fundamental difference between the likelihood model and the A-DLISTA architecture is presented in subsection 4.1. Indeed, unlike the latter, the likelihood model of VLISTA does not learn the dictionary using backpropagation. Instead, it uses the dictionary sampled from the posterior distribution.

### C.2.2  VLISTA - Posterior & Prior models

We report in Figure 6 the prior (left image) and the posterior (right image) architectures. We implement both models using an encoder-decoder scheme based on convolutional layers. The prior network comprises two convolutional layers followed by two separate branches dedicated to generating the mean and variance of the Gaussian distribution subsection 4.2. We use the dictionary sampled at the previous iteration as input for the prior. In contrast to the prior, the posterior network accepts three different quantities as input: the sensing matrix, the observations, and the reconstructed sparse vector from the previous iteration. To process the three inputs together, the posterior accounts for three separated "input" layers followed by an aggregation step. Subsequently, two branches are used to generate the mean and the standard deviation of the Gaussian distribution of the dictionary subsection 4.2.
We offer the reader a unified overview of our variational model in Figure 7.

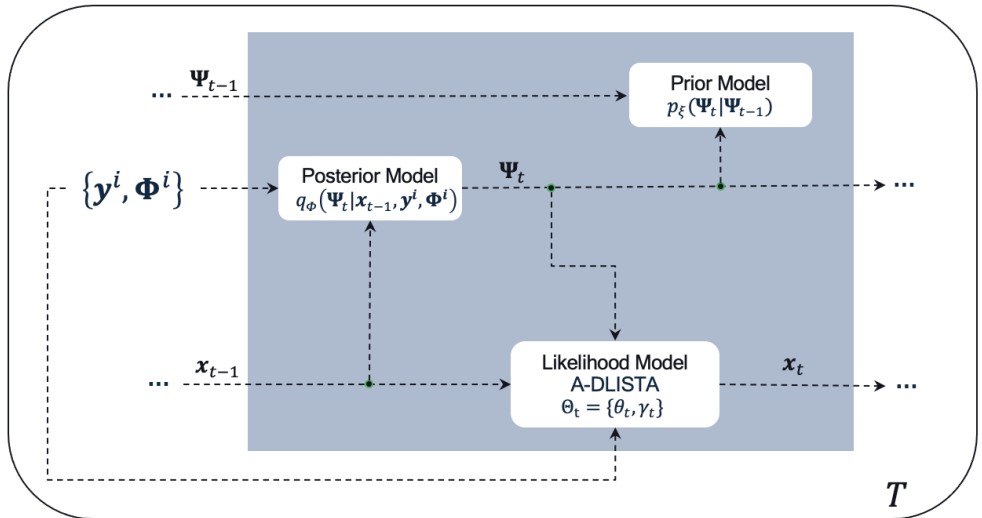

**Figure 7:** VLISTA iterations' schematic view.

## D  Training Details

This section provides details regarding the training of the A-DLISTA and VLISTA models. Using the Adam optimizer, we train the reconstruction and augmentation models for A-DLISTA jointly. We set the initial learning rate to $1.e^{-2}$ and $1.e^{-3}$ for the reconstruction and augmentation network, respectively, and we drop its value by a factor of 10 every time the loss stops improving. Additionally, we set the weight decay to $5.e^{-4}$ and the batch size to 128. We use Mean Squared Error (MSE) as the objective function for all datasets. We train all the components of VLISTA using the Adam optimizer, which is similar to A-DLISTA. We set the learning rate to $1.e^{-3}$ and drop its value by a factor of 10 every time the loss stops improving. Regarding the objective function, we maximize the ELBO and set the weight for KL divergence to $1.e^{-3}$. We report in Equation 33 the ELBO derivation.

$$\log p(\boldsymbol{x}_{1:T} = \boldsymbol{x}_{gt}^i | \boldsymbol{y}^i, \boldsymbol{\Phi}^i) = \log \int p(\boldsymbol{x}_{1:T} = \boldsymbol{x}_{gt}^i | \boldsymbol{\Psi}_{1:T}, \boldsymbol{y}^i, \boldsymbol{\Phi}^i) p(\boldsymbol{\Psi}_{1:T}) d\boldsymbol{\Psi}_{1:T} \tag{33}$$

$$= \log \int \frac{p(\boldsymbol{x}_{1:T} = \boldsymbol{x}_{gt}^i | \boldsymbol{\Psi}_{1:T}, \boldsymbol{y}^i, \boldsymbol{\Phi}^i) p(\boldsymbol{\Psi}_{1:T}) q(\boldsymbol{\Psi}_{1:T} | \boldsymbol{x}_{1:T}, \boldsymbol{y}^i, \boldsymbol{\Phi}^i)}{q(\boldsymbol{\Psi}_{1:T} | \boldsymbol{x}_{1:T}, \boldsymbol{y}^i, \boldsymbol{\Phi}^i)} d\boldsymbol{\Psi}_{1:T}$$

$$\geq \int q(\boldsymbol{\Psi}_{1:T} | \boldsymbol{x}_{1:T}, \boldsymbol{y}^i, \boldsymbol{\Phi}^i) \log \frac{p(\boldsymbol{x}_{1:T} = \boldsymbol{x}_{gt}^i | \boldsymbol{\Psi}_{1:T}, \boldsymbol{y}^i, \boldsymbol{\Phi}^i) p(\boldsymbol{\Psi}_{1:T})}{q(\boldsymbol{\Psi}_{1:T} | \boldsymbol{x}_{1:T}, \boldsymbol{y}^i, \boldsymbol{\Phi}^i)} d\boldsymbol{\Psi}_{1:T}$$

$$= \int q(\boldsymbol{\Psi}_{1:T} | \boldsymbol{x}_{1:T}, \boldsymbol{y}^i, \boldsymbol{\Phi}^i) \log p(\boldsymbol{x}_{1:T} = \boldsymbol{x}_{gt}^i | \boldsymbol{\Psi}_{1:T}, \boldsymbol{y}^i, \boldsymbol{\Phi}^i) d\boldsymbol{\Psi}_{1:T}$$

$$+ \int q(\boldsymbol{\Psi}_{1:T} | \boldsymbol{x}_{1:T}, \boldsymbol{y}^i, \boldsymbol{\Phi}^i) \log \frac{p(\boldsymbol{\Psi}_{1:T})}{q(\boldsymbol{\Psi}_{1:T} | \boldsymbol{x}_{1:T}, \boldsymbol{y}^i, \boldsymbol{\Phi}^i)} d\boldsymbol{\Psi}_{1:T}$$

$$= \sum_{t=1}^{T} \mathbb{E}_{\boldsymbol{\Psi}_{1:t} \sim q(\boldsymbol{\Psi}_{1:t} | \boldsymbol{x}_{0:t-1}, \boldsymbol{y}^i, \boldsymbol{\Phi}^i)} \left[ \log p(\boldsymbol{x}_t = \boldsymbol{x}_{gt}^i | \boldsymbol{\Psi}_{1:t}, \boldsymbol{y}^i, \boldsymbol{\Phi}^i) \right]$$

$$- \sum_{t=2}^{T} \mathbb{E}_{\boldsymbol{\Psi}_{1:t-1} \sim q(\boldsymbol{\Psi}_{1:t-1} | \boldsymbol{x}_{t-1}, \boldsymbol{y}^i, \boldsymbol{\Phi}^i)} \left[ D_{KL} \Big( q(\boldsymbol{\Psi}_t | \boldsymbol{x}_{t-1}, \boldsymbol{y}^i, \boldsymbol{\Phi}^i) \parallel p(\boldsymbol{\Psi}_t | \boldsymbol{\Psi}_{t-1}) \Big) \right]$$

$$- D_{KL} \Big( q(\boldsymbol{\Psi}_1 | \boldsymbol{x}_0) \parallel p(\boldsymbol{\Psi}_1) \Big)$$

Note that in Equation 33, we consider the same ground truth, $\boldsymbol{x}_{gt}^i$, for each iteration $t \in [1, T]$.

# E    Computational Complexity

This section provides a complexity analysis of the models utilized in our research. Table 3 displays the number of trainable parameters and average inference time for each model, while Table 4 showcases the MACs count. To better understand the quantities appearing in Table 4, we have summarized their meaning in Table 5. The average inference time was estimated by testing over 1000 batches containing 32 data points using a GeForce RTX 2080 Ti.

**Trainable Parameters and Average Inference Time.**    To compute the values in Table 3, we considered the architectures used in the main corpus of the paper, e.g. same number of layers. From Table 3, it's worth noting that although ISTA appears to have the longest inference time, that can be attributed to the cost of computing the spectral norm of the matrix $A = \Phi\Psi$. Such an operation, can consume up to 98% of the total inference time. Interestingly, neither NALISTA nor A-DLISTA require the computation of the spectral norm as they dynamically generate the *step size*. Additionally, LISTA does not require it at all. NALISTA and A-DLISTA have comparable inference times due to the similarity of their operations, whereas LISTA is the fastest model, whilst VLISTA has a higher average inference time given the use of the posterior model and the sampling procedure. Interestingly, LISTA and A-DLISTA have a comparable number of trainable parameters, while NALISTA has significantly fewer. However, it's essential to emphasize that the number of trainable parameters depends on the problem setup, such as the number of measurements and atoms. We use the same experimental setup described in the main paper, which includes 500 measurements, 1024 atoms, and three layers for each ML model. As outlined in the main paper, the likelihood model for VLISTA is similar in architecture to A-DLISTA, as reflected in the MACs count shown in Table 4. However, the likelihood model of VLISTA has a different number of trainable parameters compared to A-DLISTA. Such a dufference is due to VLISTA sampling its dictionary from the posterior rather than training it like A-DLISTA. Despite this difference, the time required for the likelihood model (shown in Table 3) is comparable to that of A-DLISTA. It's important to note that the inference time for the likelihood is reported "per iteration", so we must multiply it by the number of layers A-DLISTA uses to make a fair comparison.

**Macs count.**    Our attention now turns to the MACs count for the A-DLISTA augmentation network. As shown in Table 4, the count is upper bounded by $HWK^2 + BP$. Note that the height and width of the input are halved after each convolutional layer, while the input and output channels are always one, and the kernel size equals three for each layer (see details in Figure 5). To obtain the upper bound for the

**Table 3:** Number of trainable parameters (Millions) and Average inference time (milliseconds) for different models. Concerning the inference time, we report the average value with its error considering 10 and 50 measurements setups.

| | Parameters ($M$) | Average Inference Time ($ms$) | |
| | | meas. $= 10$ | meas. $= 500$ |
|---|---|---|---|
| ISTA | 0.00 | $54.0_{\pm0.6}$ (norm: $41.5_{\pm0.6}$) | $(1.55_{\pm0.02})e^3$ (norm.: $(1.53_{\pm0.02})e^3$) |
| NALISTA[1] | $3.33e^{-1}$ | $5.8_{\pm0.2}$ | $7.0_{\pm0.3}$ |
| LISTA | 3.15 | $1.1_{\pm0.1}$ | $1.5_{\pm0.3}$ |
| A-DLISTA[2] | 3.15 (Aug. NN: $3.11e^{-3}$) | $8.2_{\pm0.3}$ | $9.1_{\pm0.5}$ |
| VLISTA | $3.13^{\dagger}$ | $19.7^{\dagger}_{\pm0.4}$ | $21.3^{\dagger}_{\pm0.4}$ |
| VLISTA  Prior Model | 1.10 | $-$ | $-$ |
| Posterior Model | 2.08 | $3.6^{\ddagger}_{\pm0.2}$ | $4.1^{\ddagger}_{\pm0.2}$ |
| Likelihood Model | 1.05 | $2.7^{\ddagger}_{\pm0.2}$ | $3.05^{\ddagger}_{\pm0.2}$ |

[1] LSTM hidden size equal to 256;      [2] Each layer learns its own dictionary;      [†] Full model at inference - Prior model NOT used;      [‡] Single iteration.

MACs count, we set $H = \max_i(H_i) = H^{input}/2^i$ and $W = \max_i(W_i) = W^{input}/2^i$, where $H_i$ and $W_i$ are the height and width at the output of the $i$-th convolutional layer, respectively. With that in mind, we can upper bounds the MACs count for the convolutional part of the network by $HWK^2$. The convonlutional backbone is followed by two linear layers (see details in Figure 5). The first linear layer takes a vector of size $B \in \mathbb{R}^{H^{input}/16 \times W^{input}/16}$ as input and outputs a vector of length $P = 25$. Finally, this vector is fed into two heads, each generating a scalar. Therefore, the overall upper bound for the MACs count for the augmentation network is $\mathcal{O}(HWK^2 + BP + P) = \mathcal{O}(HWK^2 + P(B+1)) = \mathcal{O}(HWK^2 + BP)$, with the factor $+1$ dropped. Similar reasoning applies to the prior and posterior models of VLISTA, where we estimate the MACs count by multiplying the MACs for the most expensive layers by the total number of layers of the same type.

## F  Additional Results

In this section we report additional experimental results. In subsection F.1 we report results concerning a fix measurement setup, i.e. $\mathbf{\Phi}^i \rightarrow \mathbf{\Phi}$, while in subsection F.2 we show reconstructed images for different classical baselines.

**Table 4:** MACs count. Concerning VLISTA, we report the MACs count for each of its component: the Prior, the Posterior, and the Likelihood models.

|  |  | MACs |
|---|---|---|
|  | ISTA | $\mathcal{O}(DM(2L+N) + D^2M)$ |
|  | NALISTA | $\mathcal{O}(DM(2L+N) + [4h(d+h) + h^2]^{\dagger})$ |
|  | LISTA | $\mathcal{O}(LD(N+D) + LMN)$ |
|  | A-DLISTA | $\mathcal{O}(LDMN + [HWK^2 + BP]^{\dagger})$ |
| VLISTA | Prior Model | $\mathcal{O}(3LH_{pr}W_{pr}C_{pr}^i C_{pr}^o(K_{pr}^2 + 2T_{pr}^2)$ |
| | Posterior Model | $\mathcal{O}(L(D + M + DC_{po}^i C_{po}^o + L_{po}^i S + 10H_{po}W_{po}T_{po}^2))$ |
| | Likelihood Model | $\mathcal{O}(LDMN + [HWK^2 + BP]^{\dagger})$ |

$^{\dagger}$ Contribution from the augmentation network.

**Table 5:** Description of quantities appearing in Table 4.

| | |
|---|---|
| $M$ | Number of measurements |
| $N$ | Dimensionality of dictionary's atoms |
| $D$ | Number of atoms |
| $L$ | Number of layers |
| $h$; $d$ | Hidden and input size for the LSTM |
| $H$; $W$ | Height and width of A-DLISTA augmentation network's input |
| $K$ | Kernel size for the Conv layers of A-DLISTA augmentation network |
| $B$; $P$ | Input and output size of the linear layer of A-DLISTA augmentation network |
| $C_{po}^i$; $C_{po}^o$ | Input and output channels for the "$\mathbf{\Phi}$-input" head of the posterior model |
| $L_{po}^i$; $S$ | Posterior model bottleneck input and output sizes |
| $H_{po}$; $W_{po}$; | Height and width of posterior model's transposed convolutions input |
| $T_{po}$ | Kernel size of the posterior model's transposed convolutions |
| $H_{pr}$; $W_{pr}$ | Input and output sizes of convolutional (and transposed conv.) layers of the prior model |
| $C_{pr}^i$; $C_{pr}^o$ | Input and output channels of convolutional (and transposed conv.) layers of the prior model |
| $K_{pr}$; $T_{pr}$ | kernel size for convolutions and transpose convolutions of the prior model |

## F.1 Fixed Sensing Matrix

We provide in Table 6 and Table 7 results considering a fixed measurement scenario, i.e. using a single sensing matrix $\mathbf{\Phi}$. Comparing these results to Table 1 and Table 2, we notice the following. To begin with, LISTA and A-DLISTA perform better compared to the set up in which we use a varying sensing matrix (see section 5). We should expect such behaviour given that we simplified the problem by fixing the $\mathbf{\Phi}$ matrix. Additionally, as we mentioned in the main paper, ALISTA and NALISTA exhibit high performances (superior to other models when 300 and 500 measurements are considered). Such a result is expected, given that these two models were designed for solving inverse problems in a fixed measurement scenario. Furthermore, the results in Table 6 and Table 7 support our hypothesis that the convergence issues we observe in the varying sensing matrix setup are likely related to the "inner" optimization that ALISTA and NALISTA require to evaluate the "W" matrix.

**Table 6:** MNIST SSIM (the higher the better) for a different number of measurements with **fixed sensing matrix**, i.e., $\mathbf{\Phi}^i \to \mathbf{\Phi}$. We highlight in bold the best performance. Note that whenever there is agreement within the error for the best performances, we highlight all of them.

| | SSIM ↑ | | | | |
|---|---|---|---|---|---|
| | number of measurements | | | | |
| | 1 $(\times e^{-1})$ | 10 $(\times e^{-1})$ | 100 $(\times e^{-1})$ | 300 $(\times e^{-1})$ | 500 $(\times e^{-1})$ |
| LISTA | $\mathbf{1.34_{\pm 0.02}}$ | $3.12_{\pm 0.02}$ | $\mathbf{5.98_{\pm 0.01}}$ | $6.74_{\pm 0.01}$ | $6.96_{\pm 0.01}$ |
| ALISTA | $0.84_{\pm 0.01}$ | $0.94_{\pm 0.01}$ | $1.70_{\pm 0.01}$ | $5.71_{\pm 0.01}$ | $6.65_{\pm 0.01}$ |
| NALISTA | $0.91_{\pm 0.01}$ | $1.12_{\pm 0.01}$ | $2.46_{\pm 0.01}$ | $\mathbf{7.03_{\pm 0.01}}$ | $\mathbf{8.22_{\pm 0.02}}$ |
| A-DLISTA (our) | $1.21_{\pm 0.02}$ | $\mathbf{3.58_{\pm 0.01}}$ | $5.66_{\pm 0.01}$ | $6.47_{\pm 0.01}$ | $6.84_{\pm 0.01}$ |

**Table 7:** CIFAR10 SSIM (the higher the better) for a different number of measurements with **fixed sensing matrix**, i.e., $\mathbf{\Phi}^i \to \mathbf{\Phi}$. Note that whenever there is agreement within the error for the best performances, we highlight all of them.

| | SSIM ↑ | | | | |
|---|---|---|---|---|---|
| | number of measurements | | | | |
| | 1 $(\times e^{-1})$ | 10 $(\times e^{-1})$ | 100 $(\times e^{-1})$ | 300 $(\times e^{-1})$ | 500 $(\times e^{-1})$ |
| LISTA | $2.52_{\pm 0.01}$ | $\mathbf{3.19_{\pm 0.01}}$ | $\mathbf{4.48_{\pm 0.01}}$ | $\mathbf{6.29_{\pm 0.01}}$ | $6.74_{\pm 0.01}$ |
| ALISTA | $0.21_{\pm 0.03}$ | $0.54_{\pm 0.02}$ | $0.88_{\pm 0.01}$ | $3.54_{\pm 0.01}$ | $5.52_{\pm 0.01}$ |
| NALISTA | $1.32_{\pm 0.02}$ | $1.32_{\pm 0.02}$ | $1.06_{\pm 0.02}$ | $4.59_{\pm 0.01}$ | $\mathbf{6.88_{\pm 0.01}}$ |
| A-DLISTA (our) | $\mathbf{2.91_{\pm 0.02}}$ | $3.07_{\pm 0.01}$ | $4.26_{\pm 0.01}$ | $5.89_{\pm 0.01}$ | $6.56_{\pm 0.01}$ |

## F.2 Classical baselines

We report additional results concerning classical dictionary learning methods tested on the MNIST and CIFAR10 datasets. It is worth noting that classical baselines can reconstruct images with high quality if it is assumed that there are neither computational nor time constraints (although this would correspond to an unrealistic scenario concerning real-world applications). Therefore, while tuning hyperparameters, we consider a number of iterations up to a several thousand.

Figure 8 to Figure 13 showcase examples of reconstructed images for different baselines.

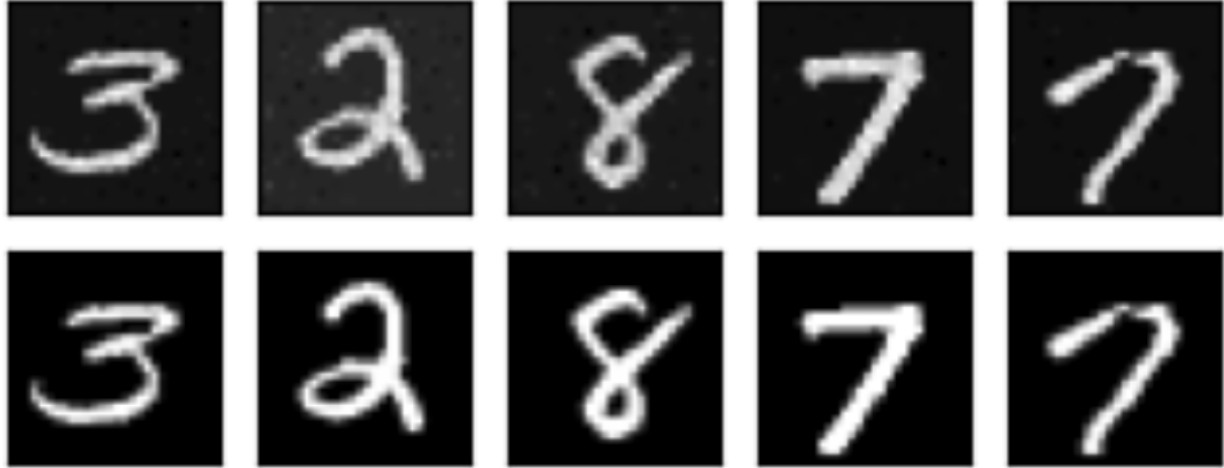

**Figure 8:** Example of reconstructed MNIST images using the canonical basis. Top row: reconstructed images. Bottom row: ground truth images. To reconstruct images we use 500 measurements and the number of layers optimized to get the best reconstruction possible.

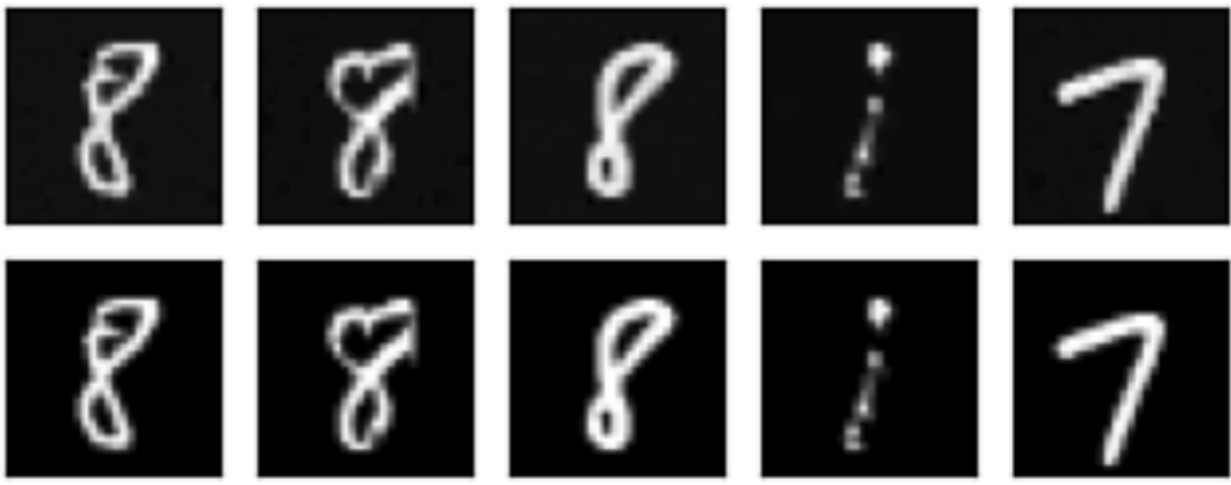

**Figure 9:** Example of reconstructed MNIST images using the wavelet basis. Top row: reconstructed images. Bottom row: ground truth images. To reconstruct images we use 500 measurements and the number of layers optimized to get the best reconstruction possible.

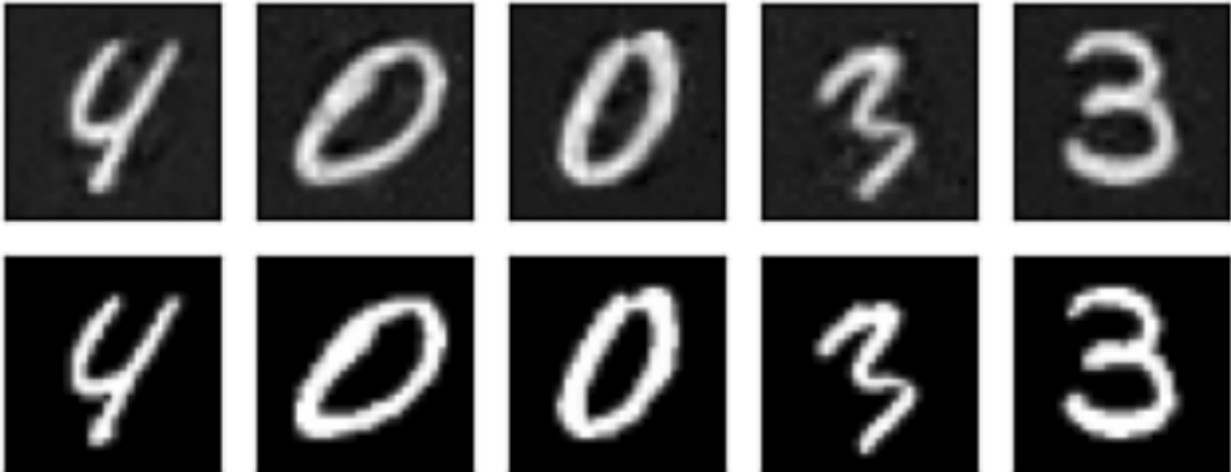

**Figure 10:** Example of reconstructed MNIST images using SPCA. Top row: reconstructed images. Bottom row: ground truth images. To reconstruct images we use 500 measurements and the number of layers optimized to get the best reconstruction possible.

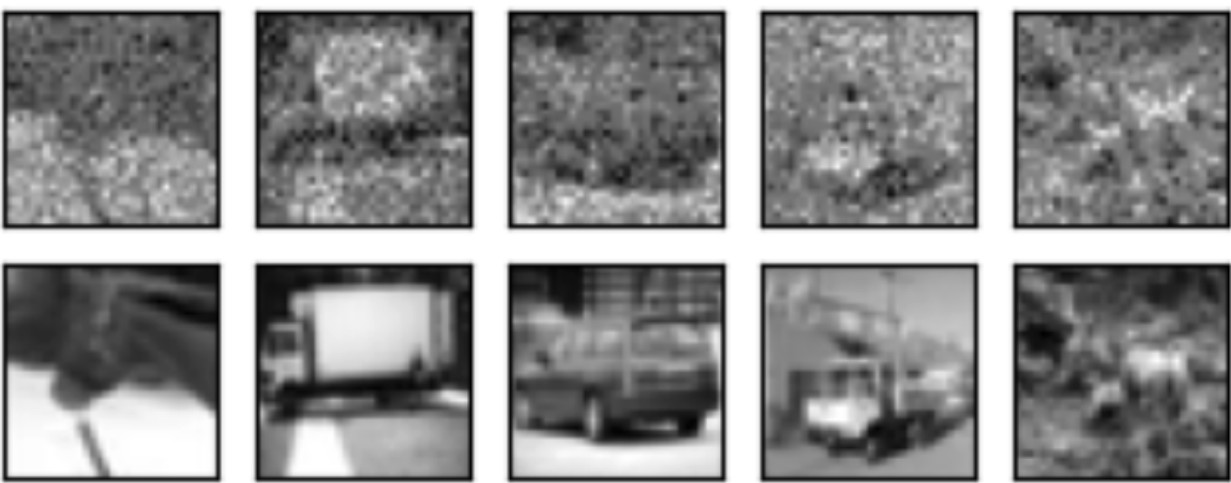

**Figure 11:** Example of reconstructed CIFAR10 images using the canonical basis. Top row: reconstructed images. Bottom row: ground truth images. To reconstruct images we use 500 measurements and the number of layers optimized to get the best reconstruction possible.

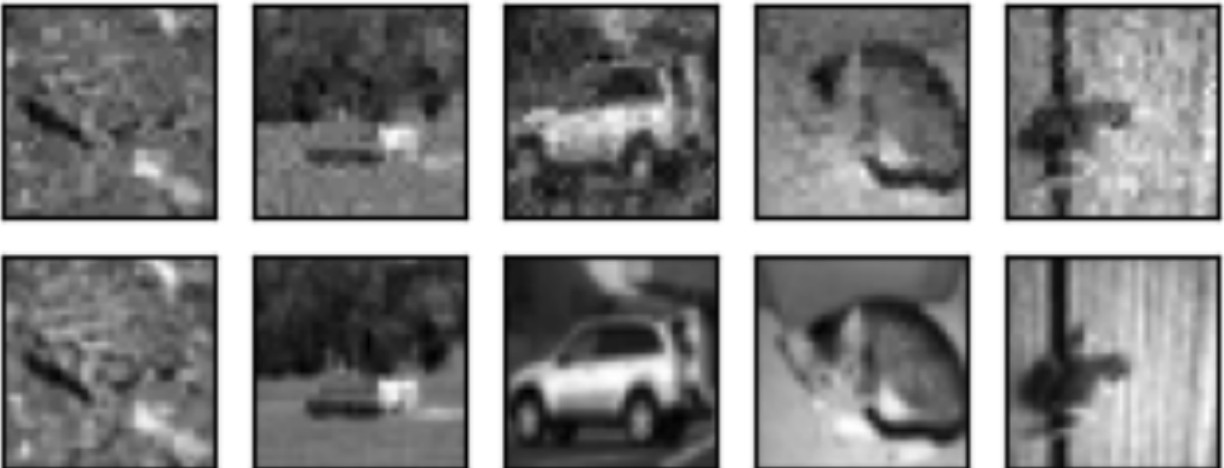

**Figure 12:** Example of reconstructed CIFAR10 images using the wavelet basis. Top row: reconstructed images. Bottom row: ground truth images. To reconstruct images we use 500 measurements and the number of layers optimized to get the best reconstruction possible.

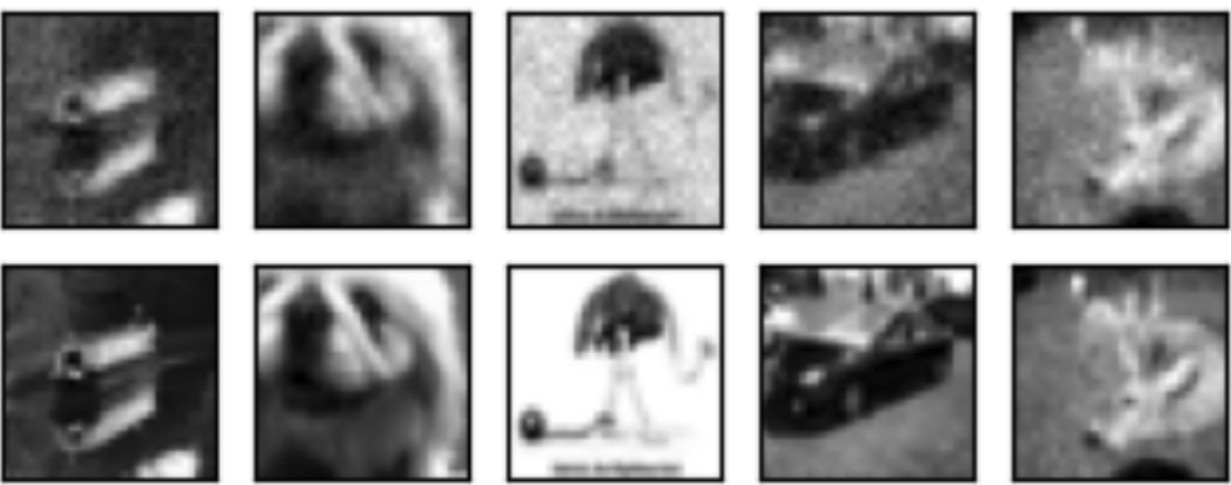

**Figure 13:** Example of reconstructed CIFAR10 images using SPCA. Top row: reconstructed images. Bottom row: ground truth images. To reconstruct images we use 500 measurements and the number of layers optimized to get the best reconstruction possible.

