# OpenReview forum: "Variational Learning ISTA"
_TMLR — Accepted by TMLR_

### Review · Reviewer_3H4v · 2024-02-05

**Summary Of Contributions:**

The author proposed a variant of the LISTA architecture. Firstly, they introduced Augmented Dictionary Learning ISTA (A-DLISTA), which includes an augmentation module to adapt to the current measurement setup. Then, they proposed a method for learning the dictionary distribution through a variational approach, named Variational Learning ISTA (VLISTA). VLISTA utilizes A-DLISTA as a likelihood model and approximates the posterior distribution of dictionaries as part of an unfolded LISTA-based recovery algorithm. Consequently, VLISTA offers a probabilistic way to jointly learn the dictionary distribution and the reconstruction algorithm under varying sensing matrix conditions. The authors provided both theoretical and experimental support, demonstrating that their model learns calibrated uncertainties.

**Audience:**

Yes

**Broader Impact Concerns:**

NA.

**Claims And Evidence:**

Yes

**Requested Changes:**

Please refer to the weaknesses.

**Strengths And Weaknesses:**

Strengths:
1. Novel Approach: The paper introduces novel methods, A-DLISTA and VLISTA, to address the issues of dictionary learning and uncertainty modeling in compressed sensing, which contribute to the existing literature in this field.

2. Theoretical and Experimental Support: The authors provide both theoretical and experimental evidence to support their proposed methods, demonstrating the effectiveness and feasibility of their approaches.

3. Calibrated Uncertainties: The paper highlights the ability of the proposed model to learn calibrated uncertainties, which is a valuable aspect in real-world applications where understanding model confidence is crucial.

Weaknesses:
1. Lack of Practical Application Context: The paper lacks concrete examples or practical application scenarios where the proposed methods could be applied. Providing such context would enhance the understanding of the practical usefulness of these techniques.

2. Limited Discussion on Computational Complexity: The paper does not extensively discuss the computational complexity of the proposed methods. Understanding the computational requirements and scalability is important for practical implementation.

3. Generalization to Wider Machine Learning Field: The paper could benefit from a discussion on how the proposed methods may generalize and contribute to the broader machine learning field beyond compressed sensing.

---

> ### Author Response · Authors · 2024-04-26
>
> We would like to thank the reviewer for all of the suggestions he made to improve the quality of our work. Upon receiving the reviewer's recommendations, we revised our manuscript to address all concerns. Our response to each of the reviewer's requests is presented in the following text.
>
> > Lack of Practical Application Context: The paper lacks concrete examples or practical application scenarios where the proposed methods could be applied. Providing such context would enhance the understanding of the practical usefulness of these techniques.
>
> We thank the reviewer for his suggestion. MRI, CT Scan imaging, and channel estimation for wireless are examples of real-world applications where the sensing matrix varies and is known. Adaptive acquisition is another example. MRI reconstruction, for example, may involve an adaptive acquisition process. In this case, the sensing matrix is sampled from a known distribution. As a result, the signal is reconstructed using the known (sampled) sensing matrix. Due to the adaptive nature of the process, each data sample is characterized by a variable but known sensing matrix [1,2]. Wireless channel estimation is another example. For example, in millimeter wave communications, the channel is sensed by beamforming codebooks. These codebooks can change in different communication rounds, resulting in a varying sensing matrix[3]. We updated our paper by adding the real-world examples. We highlighted in blue the newly added text.
>
> [1] Y. T. et al., 2021. End-to-End Sequential Sampling and Reconstruction for MRI.
>
> [2] B. T. et al., 2020. Experimental design for MRI by greedy policy search.
>
> [3] R.-F. J. et al., 2018. Frequency-domain compressive channel estimation for frequency-selective hybrid millimeter wave MIMO systems.
>
> > Limited Discussion on Computational Complexity: The paper does not extensively discuss the computational complexity of the proposed methods. Understanding the computational requirements and scalability is important for practical implementation.
>
> We thank the reviewer for his suggestion. In Appendix E (tables 3, 4, and 5) we discuss the complexity of the proposed models. We highlighted in blue the newly added text.
> Specifically, we report in Table 3 the number of trainable parameters and the average inference time concerning two different measurements setups, while in Table 4 we showcase the number of MACs for each model. Additionally, in Table 5 we describe the various quantities appearing in Table 4.
>
> > Generalization to Wider Machine Learning Field: The paper could benefit from a discussion on how the proposed methods may generalize and contribute to the broader machine learning field beyond compressed sensing.
>
> We appreciate the reviewer's comments. The work we have conducted may contribute to the literature on compressed sensing, particularly the application of LISTA-type models to solve inverse problems (characterized by underdetermined system of equations), in scenarios in which the sensing matrix changes across data points. Furthermore, our contribution may have relevance to the sparse regression literature, for which the LASSO problem is a canonical example for CS.
> This discussion has been included in the "Impact Statement" section. We highlighted in blue the text in the updated version of the manuscript.

---

### Review · Reviewer_YHuk · 2024-03-03

**Summary Of Contributions:**

The authors propose two extensions of learnable iterative soft thresholding algorithm (ISTA): (1) Augmented Dictionary Learning ISTA (A-DLISTA) that is designed to adapt with the current sample being processed, and (2) Variational Learning ISTA (VLISTA), which implements a Bayesian framework for learning a dictionary distribution that leverages A-DLISTA as the corresponding likelihood model.

My understanding of A-DLISTA in the context of its purpose is as follows. Normally, (F)ISTA, a well established convex optimization algorithm family, is applied on a sample-to-sample basis with a provided/fixed sensing and sparsification representation (everything is sample-specific, and nothing is learned/ no assumptions are made about any other samples). LISTA is the same algorithm, but with learned parameters that are intended to encode a dataset faster than ISTA. LISTA is a truncated ISTA trained to predict a fully converged ISTA that can readily be applied across a dataset. LISTA is a recurrent neural network with shared layers.

DLISTA as presented in Eqn (2) breaks the recurrence by allowing the learned matrix to be optimized for each iteration (subscript t). The learned matrix is however decomposed into two multiplicative terms via the signal model $y=\Phi\Psi x^*$. At first --because $\Phi\Psi$ always occur together-- it appears that it doesn't matter whether the $\Phi$ term is fixed since $\Psi$ is allowed to change every iteration, however I believe the fixed sensing matrix helps with convergence analysis when $\Phi$ has convenient properties.

The proposed ADLISTA has the exact same operations, but removes the remaining recurrent structure (ie it is now a fully feedforward neural network) by letting $\Phi$ also be updated by error backpropagation across samples and iterations. So the signal model has sample-specific sequence of matrices $\Phi_t$ and dataset-specific sequence of matrices $\Psi_t$. This is the meaning of the authors' term "augmented network". and "adaptivity to samples". This is central to the paper's thesis of its contributions.

VLISTA uses a variational bayesian framework to predict the dataset-specific sparsification dictionary $\Psi_t$, given $\Phi_t, x_{t-1}, y_t$, and implicitly given $\Psi_{t-1}$ (via the training procedure).

Empirical results are composed from synthetic compressed sensing scenario. MNIST, CIFAR10, and a (seemingly?) unspecified synthetic dataset are sampled by optimal compressed sensing conditions (Gaussian matrices).

**Audience:**

Yes

**Broader Impact Concerns:**

I don't see any ethical concerns .

**Claims And Evidence:**

No

**Requested Changes:**

# (a) Necessary Suggestions
1) Show the cost function you wish to optimize before presenting the recursive solution in Eqn (1). I suggest putting this on the first page because the notions of sensing matrix and sparsification dictionary/transform can easily be conflated depending on the specific field a reader is coming from. For a while I thought you were considering the case with separated synthesis/sensing and analysis/sparsifying terms in the cost function.
2) formatting:  (A) cite the (F)ista paper at Eqn(1) instead of just stating ISTA; (B) some of your quotations marks are imbalanced, I can't make it look right in MarkDown but e.g. after Eqn(1) they are curly on the left and straight on the right. (C) From Eqn 1-2 you add a subscript to sparse dictionary atoms $\Psi\rightarrow\Psi_t$ but also change the Hermetian-transpose term to regular transpose. This change should be either remedied or explained in the paper. (D) The ADLISTA and VLISTA lines look identical in Fig 3, you need to use some differentiating factor (markers, dashes, or one could be black).
3) I don't think BCS is defined?... Also, you don't explain what $x^*$ are for the synthetic case. What are these, just noise vectors? Sinusoids?
4) It's suspicious that LISTA gets better performance than ISTA, since LISTA is trained to predict converged ISTA. It shold only get the same result, faster. My guess is that you are not allowing ISTA to properly converge. The only cost functions in the paper are for VLISTA so I might just be confused. Speaking of which, I was expecting a discussion for T (number of iterations). How many are required for your method to function?
5) I apologize, but I don't really understand Figure 4 and the OOD experiments in general. Are you computing 100 p values for each noise level and each every test image and then averaging across all images and trials? I feel like a ROC curve would be more useful to understand how your algorithms fare across the test set.
6) I am having trouble understanding your generation of data. You describe picking a sample, then generating a Gaussian $\Phi_i$. Then you randomly set ~10% of $\Phi$'s entries to zero. Then your plots and tables have a "number of measurements" parameter, so here I believe you are keeping the first N rows. But what is with the 10% zeroes-- that's not enough to be considered sparse, right? 90% dense?
7) If you have analytic results that are in the appendix, you really should put the statement of the theorems in the paper at least.

# (b) Medium Suggestions
1) Motivation: You may wish to give more specific examples or higher level motivation in the introduction, i.e. to clarify what problem you are trying to solve. The purpose of a dictionary, traditionally, is to provide common representation across many samples in a dataset. In compressive sensing the role of the "dictionary" is somewhat different in that it usually is a system model or, as you put it, the sensing matrix. By giving a specific example (such as how multi-channel MRI machines have instance-specific calibration matrices as part of the system model) it could help clarify what scenario you would need a linear operator to play the role of a dictionary for a specific sample instance. [later during review]: I see you have an example on page 4; this should be in the introduction IMO. I.e. using examples to explain the problem you want to solve, not just to say what LISTA cannot do.
2) The statement "...[in LISTA] those weights [V, W] implicitly contain information about Φ and Ψ that are assumed to be known and fixed." from Sec 3.2. This is not really accuracy because they are updated during learning, I think you're talking about ISTA? Or are you referring to the fact that LISTA is trained to predict ISTA? But I don't think you did that in your experiments.
3) I think it would be really helpful to put VLISTA in the same diagram as ADLISTA. I am having a hard time understanding what a forward inference looks like, exactly, since forward inference includes backpropagation updates of $\Phi$ but also a new set of neural networks outputing $Psi$. I'm not 100% convinced this should be considered a "LISTA family" algorithm due to the latter operations.
4) Compressed sensing on MNIST in 2024 seems inappropriate. There's no written argument that can compensate for lack of realistic experimental results.

# (c) Light suggestions
1) I am satisfied with the Related Works section but [1] is another unrolled approach with sample adaptivity, but with an ADMM-family algorithm. Please consider putting an algorithm enumeration so spell out the steps. If [1] is unrelated then I do not understand your ADLISTA algorithm.
2) I would be curious to see the wallclock time during inference.


[1] Goehle, Geoff, and Benjamin Cowen. "Joint Sparse Coding and Frame Optimization." IEEE Signal Processing Letters (2024).

**Strengths And Weaknesses:**

# Strengths:
(1) The introduction and related works sections are pretty clear and well written. My only suggestion there is to add the cost function sooner and give more motivation and intuition to what you mean by sample-adaptive algorithm.
(2) I think the A-DLISTA is a good contribution. But it is not sufficiently explained (see Weakness#2)

# Weaknesses
(1) The results section is based off of pretty unrealistic toy data. The VLISTA seems like a lot of complicated machinery with a staggeringly huge number of parameters. I would never use this without a huge analysis of how prone it is to overfitting and how it performs encoding noisy /realistic data with non-Gaussian sensing matrices. Random zeros are not possible to implement in physical systems (to my knowledge) so it's unclear how well this can translate to e.g. MRI data. I'm also having a hard time understanding the presentation of OOD results (but that may be my fault).
(2) You need to spell out better what an inference looks like for ADLISTA. I imagine that first you train ADLISTA to predict ground truth (which does not exist in sparse coding...). Then, during inference, you also perform something that must still look a bit like training, i.e. joint optimization of codes and matrix coefficients. See the reference at the bottom of Requested Changes as this represents my understanding of your algorithm (something that is mixing forward-inference of the [formerly] convex algorithm with backpropagation updates of your matrices). You could hugely benefit by an enumerated algorithm box that really spells it out for us.

Most critically of all: What is the cost function for training your sample-specific $\Phi$? How many iterations of code optimization (T)? How many updates applied to the matrix coefficients in between code inferences? These details aren't addressed at all unless I missed something. These are the core reasons I have said you haven't backed up your claims but please also see Necessary Changes in the Requested Changes section.

---

> ### Author Response · Authors · 2024-04-26
> **Part 1/3**
>
> We highly appreciate the thoroughly review and want to thank the reviewer for all his effort in helping improving the quality of our manuscript. In what follows, we tried our best to address all the raised concerns.
>
> > The results section is based off of pretty unrealistic toy data. The VLISTA seems like a lot of complicated machinery with a staggeringly huge number of parameters ...
>
> In the new version of the manuscript, we discuss the complexity of the proposed models in Appendix E (tables 3, 4, and 5). We highlighted in blue the newly added text. Specifically, we report in Table 3 the number of trainable parameters and the average inference time concerning two different measurements setups, while in Table 4 we showcase the number of MACs for each model. Additionally, in Table 5 we describe the various quantities appearing in Table 4.
>
> > You need to spell out better what an inference looks like for ADLISTA ...
>
> We thank the reviewer for the suggestion. In section 4 of the new version of the manuscript we added Algorithm 1 and Algorithm 2 to show how A-DLISTA and VLISTA look like at inference.
>
> > Show the cost function you wish to optimize before presenting the recursive solution in Eqn (1) ...
>
> We thank the reviewer for his suggestions to improve our manuscript clarity. In the introduction, pages one and two, we added a brief description of the problem at hand and the objective function we want to optimize using A-DLISTA. Specifically, with A-DLISTA, we optimize a reconstruction objective similarly to other LISTA-type models. Differently, the objective for VLISTA is represented by the ELBO reported in eqn. 15. We do not report the ELBO immediately in the introduction since we believe that from the reader's perspective it is more clear to first describe the various model's components and then show how to train them.
>
> > formatting: (A) cite the (F)ista paper at Eqn(1) instead of ...
>
> We thank the reviewer for his suggestions. We modified the paper based on suggestions (A) to (D). (A) We cited Daubechies et al. (2004) and Beck \& Teboulle (2009) before Eqn 1. (B) We corrected the quotation marks. (C) The dictionary subscript has been explained and the Hermitian-transpose term has been replaced with a regular transpose since we are dealing only with real data. In the updated version of the manuscript, equations 1-2 are now equations 3-4. (D) We adjusted the plot colors.
> We highlighted the revised text in blue.
>
> > I don't think BCS is defined?... Also, you don't explain what are for the synthetic case. What are these, just noise vectors? Sinusoids?
>
> We thank the reviewer for pointing that out. We introduced BCS (Ji et al., 2008) in the related works (section 2). However, as the reviewer pointed out, we did not include the acronym in section 2, which may have caused confusion. The acronym has been added to the manuscript. Additionally, we noticed a missing citation in section 5, which we have added.
> In blue, we have highlighted the updated text.
> The description of the synthetic dataset can be found in section 5.1. Please see below the corresponding text in the manuscript: ``Concerning the latter, we follow a similar prescription as in Chen et al. (2018); Liu \& Chen (2019); Behrens et al. (2021). However, in contrast to the mentioned authors, we generate a different $\Phi$ matrix for each datum by sampling i.i.d. entries from a standard Gaussian distribution, $\Phi_{ij} \sim \mathcal{N}(0, 1/m)$, where $m$ is the number of columns of $\Phi$. To generate the ground truth sparse signals $x\in \mathbb{R}^b$ we sample the entries from a standard Gaussian as well. We set each entry to be non-zero following a Bernoulli distribution with  $p = 0.1$. We generate 5K samples and use 3K for training, 1K for model selection and 1K for testing.''
>
> > It's suspicious that LISTA gets better performance than ISTA,  ...
>
> We thank the reviewer for the interesting question. In order to resolve the recovery problem, ISTA only leverages sparsity. Conversely, LISTA may be able to leverage additional data structures through the learning process. As a result, it might be able to provide better reconstructions than ISTA.
> For each ML-based model, except for the classical baselines, we set the number of layers to three. This choice was made because we found that three layers were sufficient to discern the differences in performance between each model. Our LISTA implementation follows Liu et al. (2019), which assigns a different learnable matrix to each layer. As a result, testing each single model for each value of T would have required training tens of models. Consequently, too extensive resources were required.
> In the revised version of the manuscript, we have added Appendix E which contains a complexity analysis of the models we used, thus providing additional insight into their properties.

---

> ### Author Response · Authors · 2024-04-26
> **Part 2/3**
>
> > I apologize, but I don't really understand Figure 4 and the OOD experiments in general ...
>
> We apologize for not having explained Figure 4 clearly. In the new version of the manuscript, we have updated the text (highlighted in blue). We hope that the current explanation clarifies the figure. Regarding the reasons for using p-values, we chose them because they allowed us to draw statistically robust conclusions regarding the models' ability to reject OODs.
>
> > I am having trouble understanding your generation of data. ...
>
> It is greatly appreciated that the reviewer noticed this. This is indeed a typo on our part. To generate the synthetic dataset, we followed the same steps as Chen et al. (2018). The 10\% probability refers to the probability that the elements are not zero. The typo has been corrected and the new text has been highlighted in blue.
>
> > If you have analytic results that are in the appendix, you really should put the statement of the theorems in the paper at least.
>
> We thank the reviewer for his suggestion. Part of the analytic results have been moved from the appendix to the main text. We highlighted in blue the new text in the updated version of the manuscript.
>
> > Motivation: You may wish to give more specific examples or higher level motivation in the introduction, ...
>
> We thank the reviewer for his suggestion. We have added in the introduction a few examples of real-world applications to support our ideas in the revised version of the text. The new text has been highlighted in blue.
>
> > The statement "...[in LISTA] those weights [V, W] ...
>
> We thank the reviewer for pointing that out. Yes, that is correct. In ISTA, the matrices $\Phi$ and $\Psi$ are known and fixed. As for LISTA, the only assumption is that $\Phi$ and $\Psi$ are fixed and the weights matrices [V,W] implicitly contain information about them.
> Those parts of the text that have been corrected are highlighted in blue.
>
> > I think it would be really helpful to put VLISTA in the same diagram as ADLISTA. ...
>
> We thank the reviewer for his questions. No backpropagation updates to $\Phi$ are done in ADLISTA and VLISTA. According to Figure 1, both $y$ and $\Phi$ are input data for ADLISTA (as well as for VLISTA). In response to the reviewer's request, we modified Figure 1 to show ADLISTA and VLISTA (inference) architectures side by side.
> When it comes to ADLISTA, the dictionary is learned during training and then fixed at the inference time. Only the threshold and step size are updated by the augmentation network. In regard to VLISTA, we report the "inference" architecture. Due to the fact that the model does not use the prior model during inference, it is important to specify this detail.
> We consider VLISTA a member of the "LISTA family" given that it represents an unfolded version of ISTA. Indeed, as described in the paper, at the heart of the likelihood model there is the A-DLISTA architecture (which represents an augmented unfolded parametrized version of ISTA).
> The sampling operation of the dictionary is one of the major differences between VLISTA and all other models. While all other models use, or assume, a fixed dictionary (ISTA, LISTA, ALISTA, NALISTA) or a learned dictionary (ADLISTA), VLISTA samples a refined dictionary at each iteration. In any case, we view the dictionary sampling operation as an augmentation step, which is why we consider VLISTA to be a member of the LISTA family of models.
>
> > Compressed sensing on MNIST in 2024 seems inappropriate. There's no written argument that can compensate for lack of realistic experimental results.
>
> We would like to thank the reviewer for his comments. We present results based on three different datasets, namely MNIST, CIFAR10, and a synthetic dataset. This choice is based on the fact that these datasets offer the greatest overlap with other studies concerning LISTA-like models. As an example, the authors of NALISTA used only the synthetic dataset (the same as ours), while those of LISTA used MNIST for testing.

---

> ### Author Response · Authors · 2024-04-26
> **Part 3/3**
>
> > I am satisfied with the Related Works section but [1] is another unrolled approach ...
>
> We thank the reviewer for suggesting the recent work from Goehle G. et al. (2024) - please, note that the paper was published after our submission to TMLR. It is true that an enumeration of algorithms will assist in understanding the models we have proposed. We added algorithm enumerations in Section 4 for both the A-DLISTA and VLISTA models. As noted by the reviewer, reference [1] proposes an unrolled approach for jointly solving the sparse recovery and sparsifying transform learning problems. It should be noted, however, that the authors of [1] examined a different family of algorithms than we did. In addition, while in [1] the authors impose specific constraints on the structure of the parameterized tight frame, in our work, the model is allowed to learn the dictionary and discover any structure in the data without additional constraints.
>
> > I would be curious to see the wallclock time during inference.
>
> We thank the reviewer for his suggestion. To address such a request and the reviewer's concerns regarding the complexity of the VLISTA model, we report in tables 3-5 (Appendix E) the average inference time for each model, alongside with the number of trainable parameters and MACs count. The new text has been highlighted in blue.

---

### Review · Reviewer_8VsP · 2024-04-04

**Summary Of Contributions:**

The paper proposes two major algorithms: (1) Augmented Dictionary Learning ISTA (A-DLISTA), which uses an augmentation
module to adaptively learn the parameters according to the sensing matrix; (2) Variational Learning ISTA (VLISTA) which can learn dictionary distributions in the variational framework for compressive sensing recovery with varying sensing matrices. Numerical experiments on the three data sets have illustrated the proposed effectiveness and show that the proposed dictionary distribution learning can be applied successfully for out-of-distribution sample detection.

**Audience:**

Yes

**Broader Impact Concerns:**

The paper lacks broader impact discussions or statements, which could be further added.

**Claims And Evidence:**

Yes

**Requested Changes:**

1. Substantial language editing and proofreading should be done.
2. Theoretical and numerical justifications of prior/posterior models could be provided.

**Strengths And Weaknesses:**

Strengths: The proposed methods have certain novelty with ample compressive sensing applications, and numerical experiments seem to be convincing.
Weaknesses: The assumptions for the prior and posterior models could be further justified for real applications. Their extensions to other probability distributions could also be illustrated in more detail.

---

> ### Author Response · Authors · 2024-04-26
>
> We thank the reviewer for his help to improve the quality of our manuscript.
>
> > Substantial language editing and proofreading should be done.
>
> We tried our best to improve the writing of the manuscript.
> We updated the submitted file accordingly and we'll keep working on it if further refinement is required.
>
> > Theoretical and numerical justifications of prior/posterior models could be provided
>
> We thank the reviewer for the interesting comment. Concerning VLISTA, we use an isotropic Gaussian as a prior before the first layer only. In all the subsequent steps, the prior and posterior distributions are still Gaussian but with learnable $\mu$ and $\sigma$. Empirically, we observed that using a standard Gaussian in the initialization phase of the iterative procedure did not pose critical limitations to the performance of our model, especially because the dictionary is refined at each layer and the likelihood model adapts its parameters to it, as well as to the sensing matrix. Such a setup, allows our framework to be flexible enough to adapt to the current dataset. Moreover, although we used Gaussians to model the dictionary distribution, our framework is not at all restricted in that regard. Indeed, it can support any flexible distribution family such as mixtures of Gaussians (to get heavier tails) or even distributions arising from normalizing flows. The two necessary things are a differentiable sampling procedure (usually, defined as a means of a transformation of a base distribution) and being able to evaluate and differentiate the density of the prior and variational posterior.
>
> > The paper lacks broader impact discussions or statements, which could be further added.
>
> We added the impact statement at the end of main corpus of the manuscript after section 6. We highlighted the new text in blue.

---

### Decision · Action_Editor_co9d · 2024-06-24

**Recommendation:** Accept with minor revision

**Comment:**

The problem studied in this papers is a classic problem in signal processing: joint estimation of a signal and its sparsifying basis. That the authors also allow for changing sensing matrix in time is a nice twist adding a layer of complexity to the problem, with potential value to practitioners. The theoretical guarantees in a teacher-student set up are strengthening the  soundeness of the approches.
Many of the main reserves by reviewers were easily answered (lack of discussion on the applicability and real-case uses, on the algorithmic complexity, or, mainly, on presentation issues). The discussion has been fruitful and the revised version of the paper seem to have improved clarity. One reviewer still recommended rejection. However, the two others seem pleased by the latest paper version. My own quick reading indicates paper of sufficient quality, both in originality and clarity of exposition, to deem publication in TMLR. Also because this problem is anything but niche and may trigger the interest of part of TMLR' audience.
My main suggestion to the authors before publication would be to add a link to a working code able to reproduce the experiments.

**Audience:**

It is consensus among the reviewers that the a part of the audience of TMLR will find potential interest in this paper. In particular those interested in connections with signal processing, given the importance of compressive sensing and dictionary learning in this context. Practitioners may, too, be looking at this paper given its novel algorithmic approaches.

**Claims And Evidence:**

The paper provides both some theoretical results as well as numerical experiments on both real and synthetic data in order to back-up their newly proposed algorithms for joint learning of the sensing matrix and sparsifying dictionary parameters. The theoretical results come mainly in the form of performance guarantees (convergence to truth) in a ground-truth model. The numerical results compare various state-ot-the-art approaches on MNIST, CIFAR and synthetic data.
Despite the reserve of one reviewer, I believe that the paper provides sufficient material for the readership to make up their mind on the relevance of the proposed methods to their application domains. It is true that the datasets used for testing are relatively simple compared to current real data. Yet, the numerics follow standard benchmarks still widely used. However I did not find a link to codes. I would ask the authors to add such a link so that readers can quickly reproduce experiments.